# Meta-Transcriptomic Analysis Uncovers the Presence of Four Novel Viruses and Multiple Known Virus Genera in a Single *Hibiscus rosa-sinensis* Plant in Colombia

**DOI:** 10.3390/v16020267

**Published:** 2024-02-07

**Authors:** Avijit Roy, Sam Grinstead, Guillermo Leon Martínez, Juan Carlos Campos Pinzón, Schyler O. Nunziata, Chellappan Padmanabhan, John Hammond

**Affiliations:** 1Molecular Plant Pathology Laboratory, Beltsville Agricultural Research Center, Agricultural Research Service, United States Department of Agriculture (USDA), Beltsville, MD 20705, USA; sam.grinstead@usda.gov; 2AGROSAVIA, Centro de Investigación La Libertad, Km.17 vía Pto. Lopez, Villavicencio, Meta, Colombia; gleon31@hotmail.com (G.L.M.); jcampos@agrosavia.co (J.C.C.P.); 3Plant Pathogen Confirmatory Diagnostics Laboratory, Science and Technology, Plant Protection and Quarantine, Animal and Plant Health Inspection Service, USDA, Laurel, MD 20708, USA; schyler.o.nunziata@usda.gov (S.O.N.); chellappan.padmanabhan@usda.gov (C.P.); 4Floral and Nursery Plants Research Unit, Beltsville Agricultural Research Center, Agricultural Research Service, USDA, Beltsville, MD 20705, USA; john.hammond@usda.gov

**Keywords:** hibiscus viruses, high-throughput sequencing, carlavirus, cilevirus, potexvirus, nepovirus, soymovirus, virus discovery

## Abstract

Hibiscus is not native to Colombia but well suited to its arid soil and dry climates. A single hibiscus plant from Risaralda, showing black spots on upper and lower sides of its leaves, was collected for virome analysis using meta-transcriptomic high-throughput sequencing technology. Bioinformatic analysis identified 12.5% of the total reads in the Ribo-Zero cDNA library which mapped to viral genomes. BLAST searches revealed the presence of carlavirus, potexvirus, and of known members of the genera *Betacarmovirus*, *Cilevirus*, *Nepovirus*, and *Tobamovirus* in the sample; confirmed by RT-PCR with virus-specific primers followed by amplicon sequencing. Furthermore, in silico analysis suggested the possibility of a novel soymovirus, and a new hibiscus strain of citrus leprosis virus C2 in the mixed infection. Both RNA dependent RNA polymerase and coat protein gene sequences of the potex and carla viruses shared less than 72% nucleotide and 80% amino acid identities with any alphaflexi- and betaflexi-virus sequences available in GenBank, identifying three novel carlavirus and one potexvirus species in the *Hibiscus rosa-sinensis* plant. The detection of physalis vein necrosis nepovirus and passion fruit green spot cilevirus in hibiscus are also new reports from Colombia. Overall, the meta-transcriptome analysis identified the complex virome associated with the black spot symptoms on hibiscus leaves and demonstrated the diversity of virus genera tolerated in the mixed infection of a single *H. rosa-sinensis* plant.

## 1. Introduction

*Hibiscus* spp. are common malvaceous ornamental flowering shrubs, originally native to tropical Asia, but which grow in tropical, subtropical, and warm temperate regions anywhere in the world. Hibiscus is not native to South America, but several hibiscus species are well suited to Colombia’s arid soil and dry climates making it ideal for cultivation in gardens. It has also naturalized in tropical rainforests. Hibiscus grows well in a large range from the coast to the Andean mountains, but it appears susceptible to disease or pests on the Caribbean coast. Due to their many beneficial attributes, hibiscus flower extracts are used in medicine for the treatment of high blood pressure, as a cancer preventive, in cosmetology to prevent premature ageing and as a natural dye or skin ointments with traditional healing uses [1,2,3,4].

The viruses infecting hibiscus in Colombia are largely unexplored. Viruses known to infect hibiscus in Colombia include hibiscus chlorotic ringspot virus (HCRSV), hibiscus latent Fort Pierce virus (HLFPV), citrus leprosis virus C2 (CiLV-C2), and its two hibiscus strains CiLV-C2H and CiLV-C2H2 [5,6,7]. Very recently, hibiscus soymovirus (HSV) was detected in the Tolima, Meta, and Quindío departments (states) in Colombia [8]. Mixed infections comprising combinations of these viruses were frequently detected.

High-throughput sequencing (HTS) is a powerful tool for virus detection and the discovery of many hibiscus viruses worldwide in the last few years. So far, at least 23 viruses have been reported as infecting ornamental hibiscus worldwide, including members of the genera *Alfamovirus* (alfalfa mosaic virus) [9], *Betacarmovirus* (hibiscus chlorotic ringspot virus) [10], *Begomovirus* (cotton leaf curl Multan virus [11,12], cotton leaf curl Burewala virus [13], hibiscus golden mosaic virus [14], and hibiscus yellow vein leaf curl virus) [15], *Cilevirus* (citrus leprosis virus C, CiLV-C2) [5,6,7], hibiscus-infecting cilevirus [16], and passion fruit green spot virus (PFGSV) [17], and an unclassified cilevirus (hibiscus yellow blotch virus) [18], *Dichorhavirus* (citrus chlorotic spot virus [19], and clerodendrum chlorotic spot virus) [20], *Higrevirus* (hibiscus green spot virus 2) [21], *Nucleorhabdovirus* (eggplant mottled dwarf virus) [22], *Ilarvirus* (tobacco streak virus) [23], *Tobamovirus* (hibiscus latent Fort Pierce virus, HLFPV, [24], hibiscus latent Singapore virus, HLSV [25], hibiscus latent Hawaii virus [26], and tomato mosaic virus) [27], *Tymovirus* (okra mosaic virus) [28,29], *Tospovirus* (impatiens necrotic spot virus) [30], and *Soymovirus* (HSV) [31]. Except for the genera *Begomovirus* and *Soymovirus*, the viral genomes of all of the above-mentioned genera consist of ribonucleic acid (RNA). Although there was a report of hibiscus-latent ring spot virus (HLRSV), a possible member of the genus *Nepovirus,* infection in hibiscus from Nigeria, serological study showed no relationship between HLRSV and the recognized or possible members of the genus *Nepovirus* [32]. Moreover, no HLRSV sequence is available in GenBank to confirm the presence of a *Nepovirus* in hibiscus. A recent HTS study, on a Guangzhou *H. rosa-sinensis* sample from China, revealed the presence of nine virus species in a mixed infection [33]. Out of the nine, tobacco mild green mosaic virus (*Tobamovirus*), turnip mosaic virus (*Potyvirus*), potato virus M (*Carlavirus*), a fabavirus species (*Fabavirus*), and reads related to mitoviruses, were identified for the first time. The number of contigs and their sizes, the depth of coverage, and the actual genome coverage of each virus were not mentioned. There are no aligned retrieved sequences or raw HTS data available in GenBank specific to the Guangzhou hibiscus virus isolates. In addition, RT-PCR followed by Sanger sequencing was not performed as a confirmatory test, so the presence of these viruses detected by HTS was not confirmed. In another instance, okra yellow mosaic Mexico virus, a begomovirus associated with the yellowing of *H. sabdariffa* in Guerrero, Mexico, was reported [34] but there was no further study conducted to confirm its infectivity towards *H. rosa-sinensis* or any other hibiscus species.

The purpose of this study was to investigate the virome associated with black spots on hibiscus leaves collected from Risaralda in Colombia in 2022 utilizing a meta-transcriptomic approach. The occurrence of the known hibiscus viruses [CiLV-C2, CiLV-C2H, PFGSV, HCRSV, and HLFPV], was also addressed in this study. In addition to known hibiscus viruses, RT-PCR-specific primers were also designed to confirm the presence of the physalis vein necrosis nepovirus (PhyVNV), previously reported on the cape gooseberry (*Physalis peruviana*) in Colombia and the novel species of carla and potex virus associated with black spot disease in hibiscus.

## 2. Materials and Methods

### 2.1. Source of Hibiscus Samples

In 2020–2022, during surveys of *Brevipalpus* transmitted virus (BTV) reservoir hosts, 30 symptomatic hibiscus samples were collected from different regions in Colombia and tested for known BTVs using specific conventional RT-PCR and RT-qPCR assays [7,35,36]. All thirty samples were sent for testing to the United States Department of Agriculture-Agricultural Research Service (USDA-ARS), Molecular Plant Pathology Laboratory (MPPL), Beltsville, Maryland, United States, under an Animal and Plant Health Inspection Service—Plant protection and Quarantine (APHIS-PPQ) importation permit. About ~100 mg of the symptomatic tissue of hibiscus leaves was excised and chopped into small pieces before transfer to a 2 mL microcentrifuge tube and either processed for RNA isolation or stored at −80 °C for future use. To determine the virome involved in the mixed infection, 50% of the total samples representing the different regions of Colombia were analyzed using a meta-transcriptomic HTS (mHTS) approach. Out of 30, fifteen symptomatic hibiscus leaf samples were collected from the Andean (*n* = 6), Orinoquía (*n* = 7), and coffee growing (*n* = 2) regions. The Andean hibiscus samples, VHFR (S54) and VGH (S197), were collected from the Valle del Cauca, whereas THN (S86), THL (S91), THSA (S94), and THM (S105) were collected from the Tolima department. The samples MLH (S38), MHV (S52), MHGr (S100), MHGr3 (103), MGrH1 (S120), and MGrH (S183) were collected from Meta and sample CYH (S233) was collected from Casanare in the Orinoquía region. The remaining two hibiscus samples, QCHr (S266) and RsTHr (S269), were collected from the Quindio and Risaralda departments, in the coffee growing region. The Illumina sequencing platform was used for the mHTS study, which is a feasible method for novel virus detection in multiple samples. Interestingly, only sample RsTHr (S269), collected from Risaralda (Santa Rosa de Cabal town, 4°51.790′ N–75°36.667′ W), had black spots on the upper and lower sides of most of its leaves. The virome analysis from its HTS meta-transcriptomic data is presented in the current study. The archived HTS data of the remaining fourteen hibiscus samples from different departments in Colombia were also included in a search for newly discovered virus sequences found in the RsTHr sample.

### 2.2. RNA Isolation

Total RNA was extracted from 100 mg of insect-free leaf tissues using the RNeasy Plant Mini Kit (Qiagen, Germantown, MD, USA), following a modified instruction manual protocol. The weighed symptomatic leaf tissues were homogenized using a FastPrep FP-24 instrument (MP Biomedical, Solon, OH, USA). Malvaceous plant leaves and roots generally contain highly gummy, mucilaginous substances like polysaccharides, secondary metabolites, and, especially, polyphenolics which can interfere with downstream RT-PCR and HTS library preparation. To improve the RNA quality from hibiscus tissues during RNeasy kit extraction, an intermediate heating step at 65 °C for 3 min was added before the supernatant of the flow through from ‘lilac’ column and 96–100% of ethanol mixture was loaded into the pink RNeasy mini column. After final elution, RNA concentrations were measured, and the quality was tested with a Qubit fluorometer (Invitrogen, Carlsbad, CA, USA) and a TapeStation (Agilent, Santa Clara, CA, USA), respectively.

### 2.3. High-Throughput Sequencing and Genome Assembly Using Bioinformatics Approach

The cDNA library was constructed using the Illumina TruSeq^®^ Stranded Total RNA Library Prep Plant kit (Illumina, Inc., San Diego, CA, USA) following the modified Illumina ‘Ribo-Zero Total RNA’ recommendations [7]. The quality and quantity of the unique dual-indexed cDNA libraries were analyzed using a 4200 TapeStation instrument (Agilent, Santa Clara, CA, USA), and Qubit™ Flex Fluorometer with dsDNA High Sensitivity kit (Invitrogen by Thermo Fisher, Carlsbad, CA, USA), respectively. Pair-end sequencing was conducted at the Plant Pathogen Confirmatory Diagnostics Laboratory (PPDCL), Laurel, MD, using a NextSeq 550 system with 2 × 75 bp (V2) Hi-output sequencing reagent kits (Illumina, San Diego, CA, USA). The FASTQ sequence data files were generated by the onboard NextSeq 550 bcl2fastq2 v2.20 software and processed for bioinformatic analysis.

During bioinformatic analysis, the raw HTS reads were trimmed and filtered with Trimmomatic v0.39 [37]. Quality-trimmed reads were then assembled using SPAdes v3.15.5 [38]. Assembled contigs were searched against an in-house database comprising filtered viral protein sequences from RefSeq and the proteomes of *Arabidopsis thaliana* and *A. lyrata* (Uniprot taxon IDs 3702 and 81972). Plant-related hits including the host *H. rosa-sinensis* and *H. syriacus* genome and homologs to Arabid proteins were removed before the remaining contigs were blasted against the full NCBI nr database for confirmation. Coverage statistics were then generated for the viral contigs with bbmap v39.01 [39]. Finally, the raw data were analyzed and an output summary was created (Munging). Details of the bioinformatic pipeline are displayed in the form of a flow chart (Figure 1).

### 2.4. Sequence Analysis and Database Search

BLASTn/BLASTx/BLASTp searches were conducted using the NCBI GenBank database through their web site running on the non-redundant (nr) DNA and protein databases. Sequence identities and query coverage were also calculated using the BLAST program. The open reading frames (ORFs) and conserved domains were identified using ORF finder and the conserved domain database (CDD) at NCBI. To compare the results to the International Committee on Taxonomy of Viruses (ICTV)-approved taxonomic demarcation criteria, pairwise sequence comparisons were computed using multiple independent pairwise alignments in the Sequence Demarcation Tool (SDT) ver. 1.3 software [40]. Moreover, to further aid in the classification of sequences, the SDT-produced quality pairwise identity plots and color-coded distance matrices were compared to the appropriate demarcation criteria to determine whether the sequences fell within existing species boundaries or should be recognized as novel species.

### 2.5. Phylogenetic and Recombination Analyses

Phylogeny was approximated using the multiple sequence alignment program MAFFT (Multiple Alignment using Fast Fourier Transform) [41] and RAxML-NG (Randomized Axelerated Maximum Likelihood—Next Generation), a popular tree inference tool to build ML trees. It combines the strengths and concepts of the previously developed RAxML and ExaML (Exascale Maximum Likelihood) [42,43] and supports all 22 ‘classical’ General Time Reversible (GTR)-derived models. Alignments were performed using MAFFT v7.520 using the global pair strategy [44]. ModelTest-NG v0.1.7 was used to select the best-fitting model, and RAxML-NG v1.2.0 was then used to build ML trees using 1000 bootstrap replicates [45]. Bootstrap convergence was determined using autoMRE cutoff 0.03. All novel carlavirus sequences were run using protein alignments of their polyprotein RNA-dependent RNA polymerase (RdRp), Triple gene Block (TGB) 1, TGB2, TGB3, and CPG with 46 RdRp, 45 TGB1, 40 TGB2, 35 TGB3, and 24 CPG reference protein sequences from the RefSeq database. The polyprotein and coat protein gene (CPG) sequences of partial hibiscus carlavirus were truncated by gaps in the assembly. The potexvirus was run with 76 other potexviruses aligning complete genome sequences. The nepovirus was run with 37 RNA1 and 38 RNA2 genome sequences of other nepoviruses and in protein alignments with the polyprotein ORFs from both the segments.

To identify possible recombinant events among the hibiscus carlaviruses (HiCaVs), the full genome sequences of 64 carlavirus species were included along with complete HiCaV genome sequences. Incomplete carlavirus genome sequences were not included in recombination analysis. Monopartite carlavirus sequences were aligned using MAFFT [41]. Recombination events were detected using several recombination analysis programs built into the Recombination Detection Program (RDP v4.101) with default parameter values [46]. A cutoff of at least five different methods detecting the same event at *p*-value < 0.05 was used to determine true positives [47].

### 2.6. Viral Genome Sequence Detection and Validation

To validate the identified viral sequences in the hibiscus sample, primer pairs were designed using the assembled virus-annotated contigs (Table 1). In addition, previously published primers for detection of CiLV-C2, CiLV-C2H, HSV, and PFGSV [6,8,31,48,49] were also included for testing (Appendix A). All the reverse primers were combined and a 10× primer mix containing 2 µM of each reverse primer was prepared. The first-strand cDNAs were synthesized using 10 µL of total RNA @100 ng/µL, 2 µM of virus-specific reverse primer mix (5.0 µL), 10 mM of dNTP (2.5 µL), 12.75 µL of nuclease-free water, 10 µL of 5× first-strand buffer, 5 µL of 0.1 M dithiothreitol (DTT), 2.5 µL of RNaseOUT™, and 2.25 µL of Superscript III Reverse Transcriptase (Invitrogen, Carlsbad, CA, USA). Tubes each containing 50 µL of reaction mix were incubated at 50 °C for 50 min followed by 72 °C for 15 min in a thermal cycler to inactivate the cDNA reaction. Transcribed cDNA (2 µL) was used as a template for separate 25 µL PCR reaction mixes containing 12.5 µL of 2× Green GoTaq^®^ Reaction Buffer (3 mm MgCl_2_, 400 µM dNTP) (Promega, Madison, WI, USA), each with 1 µL of appropriate forward and reverse virus-specific primers with a final primer concentration of 0.4 µM (Table 1) and 10.5 µL of make-up volume of nuclease-free water. The PCR amplicons were excised from the agarose gel and cleaned using the QIAquick Gel Extraction Kit (Qiagen, Germantown, MD, USA) and sequences were confirmed through Sanger sequencing in both directions using virus-specific primer pairs.

## 3. Results

### 3.1. Symptom Observation on Hibiscus Leaves

During BTV-field surveys conducted in citrus growing regions in Colombia during 2020–2022, a total of 30 hibiscus samples showing green ringspots with central chlorotic spots in senescing areas, mosaic, and black or chlorotic spot symptoms were collected. Natural cilevirus infections (CiLV-C, CiLV-C2, CiLV-C2H, and PFGSV) produced green ringspots with internal chlorotic spots in senescing *H. rosa-sinensis* leaves or irregular green rings, necrotic ringspots, or chlorotic round lesions with green borders were observed [7,17]. A hibiscus sample (RsTHr), having an unusual black spot symptom on the upper and lower sides of the leaves (Figure 2), was collected from Risaralda, a department of Colombia, and selected for virome analysis following a meta-transcriptomic approach. More severe symptoms were observed on the upper surface of the leaves where several black spots had coalesced to form a larger irregular spot.

### 3.2. Detection and Discovery of Hibiscus Viruses Using HTS and Read Assembly

Identification of potential viruses in symptomatic *H. rosa-sinensis* sample RsTHr was determined using meta-transcriptomic analysis. A RiboZero cDNA library of the RsTHr hibiscus sample was run in the NextSeq Illumina platform. During the process, the post-trimmed sequences were mapped to the Arabid proteome and the available hibiscus genome sequences in the NCBI database and the identified host sequences were then removed. The remaining unmapped reads (12.5% of total reads) were used for de novo assembly to map against plant viruses. The distribution of HTS reads from RsTHr hibiscus sample Ribo-Zero cDNA library was recorded (Figure 3). In total, 160 plant virus-related contigs were obtained, with a maximum size of 8229 nt and a minimum of 71 nt. The assembled contigs were blasted against the NCBI database for virus identification. Finally, the 116 edited and combined contigs were annotated as potential plant viral sequences, of which one contig each of betacarmovirus (3931 nts) and potexvirus (6426 nts), two contigs each of blunervirus (202 and 250 nts) and nepovirus (3901 and 7125 nts), three contigs of soymovirus (260–1250 nts), 11 contigs of mycoviruses (191–700 nts), 13 contigs of tobamovirus (194–758 nts), 18 contigs related to carlaviruses, and 65 contigs representing cileviruses were assembled. The results of the reads analysis are summarized in Table 2. The combined method of subtraction of hibiscus host sequences followed by the assembly of reads acquired from the RsTHr meta-transcriptomic library was used to deduce the genome sequence of known and novel viruses in the hibiscus sample. In total, infections by members of twelve plant virus genera were identified in a single hibiscus plant. BLASTx searches of 18 assembled carlavirus contigs revealed the presence of three nearly complete and one partial genome sequence of novel carlaviruses, whereas 65 contigs of cileviruses represent the almost complete genome sequences of three known cileviruses (CiLV-C2, CiLV-C2H and PFGSV) in the mixed infection.

The betacarmovirus, cilevirus, nepovirus, soymovirus, tobamovirus, and mycoviruses identified in the RsTHr sample by bioinformatic analysis belong to genera previously known to infect hibiscus. Two large contigs of nepovirus RNA1 and RNA2 were assembled which shared more than a 90% nt identity with the PhyVNV genome. In addition, two small contigs related to blunervirus may represent the first appearance of this virus genus in hibiscus, but further confirmation is required to substantiate this finding. In this study, we identified three nearly complete and another significant partial genome that are distinct from each other and from previously characterized carlaviruses. All four novel carlavirus sequences shared less than 70% nucleotide and 50% amino acid identities with virus sequences available in GenBank and were tentatively named hibiscus carlavirus (HiCaV)-A, HiCaV-B, HiCaV-C, and HiCaV-D, respectively. Similarly, no potexvirus infections had previously been reported from hibiscus, but we recovered a complete potexvirus genome most closely related to physalis virus X (Table 2). However, the potexvirus sequence shared less than 70% nucleotide and 55% amino acid identities with available virus sequences and was tentatively named hibiscus virus X (HiVX).

### 3.3. Confirmation of the Presence of Viruses in Infected Hibiscus by RT-PCR

To verify the presence of viruses detected by bioinformatic analysis in RsTHr, newly designed primers from the virus-annotated contigs as well as previously published primers were used for RT-PCR amplifications (Table 1 and Appendix A). Two pairs of primers of each novel virus species as well as for the new PhyVNV variant infecting hibiscus (PhyVNV-Hib) were used in RT-PCR assays and each amplicon was sequenced to confirm viral identity. The primer pairs HiCaV-A-1F/-1R and HiCaV-A-2F/-2R designed from the RNA dependent RNA polymerase (RdRp) region (ORF1), successfully amplified the expected fragments of 434 bp and 707 bp of HiCaV-A, respectively. These amplicons shared maximum 65% and 72% nt sequence identities to apple stem pitting virus (MZ147976.1) and Asian prunus virus 3 (KR998050.1) foveavirus with genome coverage of 79% and 16%, respectively. In contrast, the amino acid sequences of those amplicons covered 98–99% of the translated protein and shared maximum 59% and 53% aa identity with elderberry carlavirus E (YP_009224952.1) and ligustrum virus A (YP_009288956.1), respectively. The primer pairs HiCaV-B-1F/-1R and HiCaV-B-2F/-2R were used to amplify the partial RdRp and the overlapping region of TGB1 (ORF2) and TGB2 (ORF3) of HiCaV-B with expected amplicon sizes of 761 bp and 702 bp, respectively. Even though the 702 bp amplicon had 59% aa identity with 38% query coverage to sweet potato C6 virus (SPC6V) (YP_006589921.1), or 44% aa identity with 63% query coverage to the chrysanthemum virus B (CAE54958.2), the 761 bp amplicon did not show any significant nucleotide or translated protein similarity with current sequence data at NCBI (Table 1). The HiCaV-C was detected by amplifying 711 and 675 bp fragments using primer pairs HiCaV-C-1F/-1R and HiCaV-C-2F/-2R corresponding to RdRp and the overlapping region of RdRp and TGB1, respectively. The 711 and 675 bp amplicon sequences had 49% and 77% aa identities to SPC6V (YP_006589919.1) with 94% and 32% amplicon coverage, respectively. Increased query coverage (94 to 98% and 32 to 33%) for the 711 and 675 bp amplicons, respectively, decreased (49 to 41% and 77 to 72%) aa identities to other carlaviruses (Table 1). It was also observed that the 675 bp amplicon sequence shared 72% nucleotide identity with 33% amplicon coverage to cowpea mild mottle virus (OP611426.1). The HiCaV-D was detected by amplifying 696 and 464 bp fragments of the RdRp region using primer pairs HiCaV-D-1F/-1R and HiCaV-D-2F/-3R; these two amplicons shared only 8–15% of the amplicon size with 100% nt identity with SPC6V (NC_018448.1), whereas increased amplicon coverage at amino acid level (89–99%) reduced the identities to 35–46% to other carlaviruses (Table 1). HiVX was detected by amplifying 724 and 701 bp fragments using the primer pair HiVX-RdRp-F/R and HiVX-CPG-F/R targeting the RdRp and coat protein genes, respectively. The RdRp and CPG amplicon amino acid sequences are up to 51–55% identical to those of the agave potexvirus 1 with 59% and 99% coverage of the CPG and RdRp amplicons, respectively. In contrast, the nucleotide sequences of the RdRp and CPG amplicons shared 66–72% nt identities with papaya mosaic virus (NC_001748.1), plantago asiatica mosaic virus (KU697313.1), agave potexvirus 1 (MW328740.1), and yucca alphaflexivirus 1 (MW328742.1), respectively, covering only 30–42% of the amplicon sequences (Table 1). The nepovirus PhyVNV was detected by amplifying 828 and 651 bp fragments using the primer pair HiNeV-RNA1-F/R and HiNeV-RNA2-F/R, targeting the RNA1 and RNA2 genomic sequences, respectively. The amplicons share 93–94% nt identity (100% genome coverage) with the RNA1 and RNA2 of PhyVNV isolated from *P. peruviana* in Antioquia, Colombia (OM897214-OM897215).

The primer pairs HCRSV-F/-R and HLFPV-CPG-1F/-1R were designed and utilized to amplify the partial CPG of HCRSV and of HLFPV with amplicon sizes of 765 bp and 442 bp, respectively. These amplicons shared 97% and 99% nt identities with HCRSV isolate SBO1 (MK27967.1) and HLFPV isolate AV (MT942636.1), respectively. The primer pairs C13F/C13R and C6F/C6R [48] were used for amplification of 321 and 244 bp amplicons of PFGSV-RNA1, whereas primer pair C8F/C8R successfully amplified a 299 bp amplicon of PFGSV-RNA2. The primer pairs SVF1/SVR1 [31], HSV-REP-F/-R and HSV-CPG-F/-R [8] were designed to amplify the partial ORF VI, REP, and CPG of HSV with expected amplicon sizes of 430, 631 and 401 bp, respectively, but did not yield any amplicons. A CiLV-C2H-specific 1447 nt amplicon was obtained using the primer pair R1CiLV-C2-GF5/R1CiLV-C2H-R5 targeting the overlapping region of RdRp and CPG but no CiLV-C2 amplicon was observed using the CPG gene-specific primer pair CiLV-C2-CPG-F/CiLV-C2-CPG-R. In conclusion, four possible novel carlaviruses (HiCaV-A, HiCaV-B, HiCaV-C, and HiCaV-D) and a novel potexvirus (HiVX) were discovered and validated in the virome of hibiscus sample RsTHr. In addition, five known RNA viruses, HCRSV, HLFPV, PFGSV, PhyVNV, and CiLV-C2H (hibiscus strain of CiLV-C2), were also validated as infecting hibiscus in Risaralda, Colombia, using the respective virus-specific PCR assays, but the presence of HSV was not validated.

### 3.4. Genomic Organization of Novel Carlaviruses

Carlavirus contigs were detected in the RsTHr library and the total carlavirus reads (215,374) accounted for the third-highest proportion of the virus-related reads (10.24%) after betacarmovirus and nepovirus-related reads (Table 2). The obtained carlavirus reads were further categorized within the four possible novel species at the rate of 45.80 (HiCaV-A), 6.93 (HiCaV-B), 46.08 (HiCaV-C), and 1.31 (HiCaV-D) percentages of total carlavirus reads. The genomic sequences of HiCaV-A, HiCaV-B, and HiCaV-C were 8229, 8202, and 8117 bp long, respectively, excluding the poly(A) tail at the 3′ end. BLASTn analysis of hibiscus carlaviruses (HiCaVs) revealed 67.62–70.06%, 67.11–71.22%, and 67.90–70.52% localized nt sequence identity of HiCaV-A, HiCaV-B, and HiCaV-C, respectively, with the highest genome coverage of 16–20%, 15–18%, and 17–22% to other carlavirus genomes in NCBI. The *Carlavirus* type member, carnation latent virus (MN450069.1) genome, contains six open reading frames (ORFs) whereas the HiCaVs genomes contain five definite ORFs, with ORF1 encoding viral replicase protein, ORFs 2, 3, and 4 encoding the conserved TGB proteins, and ORF5 encoding a coat protein. No predicted ORF6 coding for a cysteine-rich protein (CRP) was found in their genome structure (Figure 4) [50]. Even though the untranslated regions (UTRs) were not determined using rapid amplification of cDNA ends (RACE), the assembled contigs of HiCaVs revealed a maximum of 77 nt at the 5′UTR and 98 nt following the end of the CPG at the 3′ end.

The complete genome sequences of HiCaV-A, HiCaV-B, and HiCaV-C, and the concatenated nucleotide sequence of HiCaV-D, were compared with sequences available in the public database, using BLASTn. HiCaV-A, HiCaV-B, and HiCaV-C shared 68.01–68.77%, 67.37–68.28%, and 66.67–66.73% nt identities with 20–22%, 18–20%, and 17–18% genome coverage with SPC6V (NC_018448), Beijing sediment betaflexivirus (MW897314), and elm carlavirus (LT898349), respectively (Appendix A). In addition, other viruses listed in the Appendix A also shared 67–72% nt identity with 10–22% query coverage only. Apart from the three novel carlavirus genome sequences, four more distinct contigs (836, 3009, 641, and 1899 nt), identified as carlavirus sequences, were also revealed, and tentatively named HiCaV-D. In addition to missing about 103 nt at the 5′ end of the RdRp ORF, gaps of 106 and 640 nt were also identified between the contigs of 836 and 3009 nt and the contigs of 3009 and 641 nt, respectively, when aligned and superimposed with the most closely related HiCaV-C sequence (Figure 4, Table 3). Attempts to amplify the gap regions between these HiCaV-D contigs were not successful. The four contigs had maximum 76.38%, 69.40%, 67.83%, and 73.60% nt identities with 29%, 4%, 97%, and 17% coverage with hydrangea chlorotic mottle virus (NC_012869), apple stem pitting virus (LC533839), Beijing sediment betaflexivirus (MW897314), and butterbur mosaic virus (ButMV) (OK432518), respectively. HiCaV-A, HiCaV-B, and HiCaV-C shared 68.96–72.33% nt identity with 77–91% genome coverage with each other, whereas the concatenated HiCaV-D sequence shared 76.16%, 67.91%, and 88.14% nt identity with HiCaV-A, HiCaV-B, and HiCaV-C, respectively, with 53–77% genome coverage (Table 4). This suggests that HiCaV-D may be a divergent variant of HiCaV-C.

The RdRp protein of HiCaV-A shared 52.83% aa identity with 81% query coverage with SPC6V (YP_006589919.1), whereas 42.81–43.62% aa identities with 99% query coverage were detected to American hop latent virus (ALJ56053.1) and poplar mosaic virus (PopMV) (WBG54312). BLASTp analysis revealed that HiCaV-B RdRp had 45.12%, 51.15%, and 51.24% aa sequence identities to SPC6V (YP_006589919.1), rose virus B (WCC72446.1), and elderberry carlavirus E (YP_009224952.1) with the highest query coverage of 100, 79, and 78%, respectively. HiCaV-C RdRp shared 43.94 and 43.47% aa identities with 99 and 100% query coverage with PopMV (WBG54312), and elderberry carlavirus E (YP_009224952.1), respectively (Figure 5A and Appendix A). The concatenated RdRp protein of HiCaV-D shared 91.12% and 61.01% amino acid identity with 78% query coverage with HiCaV-C and was most closely related to the GenBank sequence of SPC6V (YP_006589919.1), respectively (Figure 5A). The RdRp proteins of the four HiCaVs shared minimum 60.39% and maximum 91.21% aa identities between HiCaV-B and -D and HiCaV-C and -D, respectively, with 78% query coverage (Table 4). CDD analysis revealed six putative conserved domains: Betaflexiviridae_RdRp (accession; cd23245); Vmethyltransf (accession; pfam01660); RdRp_2 (accession; pfam00978); OTU_RDRP-like (accession; cd22792); Peptidase_C23 (accession; pfam05379); and viral_helicase1 (accession; pfam01443), in the RdRp protein among all HiCaVs. In addition, an extra domain, HemD (accession; cd06578), covering aa residues 678–762 of the RdRp protein was detected only in HiCaV-A.

The length of the translated CPG amino acids of HiCaV-A (308 aa), HiCaV-B (313 aa), and HiCaV-C (308 aa) varies. The CPGs of these three HiCaVs shared 46.84–52.14% and 47.39–50.52% aa identities with 82–96% and 89–99% query coverage with SPC6V (YP_006589923.1) and the helleborus net necrosis virus (HeNNV, ACM45995.1), respectively (Figure 5B). The partial CPG sequence of HiCaV-D (444 nt) shared minimum 73.33% and maximum 92.57% nt identity with 100% query coverage with HiCaV-A and HiCaV-C, respectively. The 148 aa translated from the start codon of the CPG of the four HiCaVs shared a minimum 59.57% (HiCaV-B vs. HiCaV-D) and maximum 94.59% (HiCaV-A vs. HiCaV-C) amino acid identity with 45 and 48% genome coverage, respectively (Table 4). In contrast, HiCaV-D and GenBank-deposited carlavirus sequences shared only 31.43% and 40.40% amino acid identity with 91% and 66% query coverage (148 aa) with SPC6V (YP_006589923.1) and HeNNV (ACM45995.1), respectively (Figure 5B).

The length of the HiCaV-D TGB1 is 6 aa shorter (225 aa) than the other 3 HiCaV TGB1 protein (Table 3). All four HiCaVs (HiCaV-A, -B, -C, and -D) TGB1 proteins shared 36.66–38.96%, 35.50–37.99%, 38.82–41.77%, and 36.32–38.98% aa identities and 100% query coverage with garlic common latent virus (GCLV) (QED43240.1), garlic latent virus (GLV) (QED43588.1), elm carlavirus (SNQ27852.1/SNQ27847), and chrysanthemum virus B (UZP17216.1/CAE76641), respectively (Appendix A). In addition, cowpea mild mottle virus (UDF48730.1) also aligned with 98% of the query sequence and shared 35.62–37.55% aa identity with HiCaVs. TGB1 of the four HiCaVs shared a minimum 61.04% between HiCaV-B and HiCaV-D, whereas maximum 87.88% amino acid identity was obtained between HiCaV-C and HiCaV-D (Table 4).

The TGB2 protein sequences of HiCaVs also vary in length (106 or 107 aa) (Table 3). The query sequence of the complete TGB2 protein of all HiCaVs shared aa identity @ 53.64–56.88%, 49.53–55.05%, 53.21–58.18%, 50.93–55.14%, and 49.09–54.55% with GLV (CAA92817.1, QED43853), GCLV (QED43133.1), shallot latent virus (BAZ96205.1, QYA72386), ButMV (YP_003324583.1), and helleborus mosaic virus (YP_009664742.1), respectively, with 100% query sequence coverage (Appendix A). In addition, the genome coverage and aa identity of all the HiCaVs with other distant relatives (SPC6V, elm carlavirus, and apple stem pitting foveavirus) are included in Appendix A. TGB2-translated proteins of HiCaVs also shared minimum 64.49% and maximum 97.17% aa identity between the four HiCaVs (Table 4).

Like the other ORFs, the length of TGB3 also varies, from 64 to 68 aa (Table 3). The HiCaV-A TGB3 shared 42.37% aa identity with 89% query coverage with potato virus S (PVS) sequences in the BLASTn analysis, whereas maximum query coverage (93%) shared 40.62% aa identity with passiflora latent virus (QWT83588.1) and hydrangea chlorotic mottle virus (ABD95728.1) (Appendix A). The HiCaV-B shared 39–46% aa identities with 94–98%, TGB3 query coverage with cherry green ring mottle virus (ALP45874.1), and GLV (AFV36807.1), respectively (Appendix A). BLASTp analysis revealed that HiCaV-C had full query coverage only with potato virus P (YP_001429588.1), sharing 39.06% aa sequence identity. Moreover, HiCaV-C TGB3 also shared 39–40% aa identities with 96–98% query coverage with GLV (AFV36807.1) and PVS (QWT83669.1), respectively (Appendix A). HiCaV-D shared maximum 96% query coverage with 39–42% aa identities with GLV (AFV36807.1) and PVS (QWT83669.1), but the same coverage provides the highest e-value (3 × 10^−4^) with 40% aa identity to banana mild mosaic virus (WAB21370). Interestingly, Sichuan betaflexivirus 1 (QYW08494.1) shared 33.82–41.94% aa identity with 93–96% query coverage with all the HiCaVs (Appendix A). Interspecies comparisons between the HiCaVs show 47.83–90.62% amino acid identities between the TGB3 protein sequences, with 63–100% coverage of ORF4 (Table 4).

To be considered a member of the genus *Carlavirus*, a new virus species should possess less than 72% nt and less than 80% aa sequence identity to all known viruses in its RdRp or CPG (https://ictv.global/report_9th/RNApos/*Betaflexiviridae,* accessed on 8 January 2024). Since the nucleotide as well as amino acid sequence identity of the RdRp and CPG of HiCaV-A, and HiCaV-B, and RdRp of HiCaV-C meet the threshold for species demarcation [47] (Table 4), they can be recognized as new species of the genus *Carlavirus*. The concatenated RdRp and CPG sequence of HiCaV-D shared 86.47–92.57% nt and 91.21–94.59% aa with 78 and 48% ORF coverage, respectively, with HiCaV-C. So, tentatively, HiCaV-D is designated as a divergent strain of HiCaV-C. The HiCaV-A, HiCaV-B, HiCaV-C, and HiCaV-D genome sequences were deposited in GenBank under accession numbers PP115947, PP115948, PP115949, and PP115956-59, respectively.

### 3.5. Phylogenetic Relationship and Recombination Analysis of Hibiscus Carlaviruses

Phylogenetic trees were constructed utilizing the amino acid sequences of the RdRp, TGB1, TGB2, TGB3, and the CPG of HiCaV-A, HiCaV-B, HiCaV-C, HiCaV-D, and the sequences of closely and distantly related carlavirus species available in the NCBI database (Figure 5C,D and Appendix A). Due to lower amino acid identity with known carlaviruses in GenBank, closely related foveaviruses were also included in the phylogenetic analysis. All the HiCaVs clustered together irrespective of the ORF region sequence considered in establishing the phylogenetic trees. The phylogenetic relationship based on RdRp and CPG revealed that HiCaVs clusters were closely related with SPC6V by sharing 43–44% and 41–52% aa identities, respectively (YP_006589919.1 and YP_006589923.1) (Figure 5C,D). Apart from SPC6V, all the HiCaVs shared closest distances (Figure 5A,B) with the foveaviruses in the adjacent clade of the phylogenetic tree generated (Figure 5C,D).

In contrast, the TGB1 sequences of the HiCaVs cluster shared closest distances with yam virus Y (QCT25580) and dioscorea virus A (BBJ21447). The TGB2 sequences were most closely related to ButMV (YP_003324583), whereas the TGB3 sequences clustered with grapevine rupestris stem pitting associated virus (QCY41315) followed by banana mild mosaic virus (NP_112032) (Appendix A). Moreover, HiCaV-TGBs amino acid sequences also shared the closest distance with the same viruses displayed in the phylogenetic trees (Appendix A). The combined phylogenetic analysis results demonstrate that HiCaV-A, HiCaV-B, and HiCaV-C meet the criteria to represent three new members of the genus *Carlavirus*.

To determine whether the new HiCaVs are novel species or recombinants, the possibility of recombination event(s) in the HiCaV genome sequences was investigated using RDP4 (v4.101). Only four out of nine built-in recombination detection programs in RDP4 detected the recombination events (*p*-value < 0.05) almost in the same position within the HiCaV-A, HiCaV-B, and HiCaV-C carlaviruses, but these putative inter-HiCaV recombinations might not have caused this signal (Table 5). Thus, HiCaVs might have a recombination event with other carlaviruses but no positive recombination event was confirmed within the HiCaV genomes (Figure 6).

### 3.6. Sequence Assembly, Genome Organization, and the Phylogenetic Relationship of the Potexvirus Identified in Hibiscus

A total of 153,362 processed reads related to potexviruses were detected in the RsTHr library, which is the fourth highest by percentage of viral reads (7.29%) after betacarmo-, nepovirus-, and carlavirus-related reads (Table 2). The genomic sequence of HiVX was 6406 bp long, excluding the poly(A) tail. The assembled single long contig of HiVX revealed a 93 and 131 nt long 5′- and 3′-UTR, respectively. BLASTn analysis of HiVX revealed 64.85–65.80% nt sequence identity with 24–38% genome coverage with strawberry mild yellow edge virus (SMYEV) (MZ326676), agave potexvirus 1 (MW328740), and yucca alphaflexivirus 1 (MW328742), respectively (Appendix A). The HiVX genome is organized into five ORFs similar to other potexviruses and several currently unassigned alphaflexiviruses. ORF1 (nt 94–4170) encodes the RdRp, consisting of 1358 aa, followed by three partially overlapping ORFs: ORF-2 (nt 4205–4870), -3 (nt 4860–5252), and -4 (nt 5122–5312) encode the conserved TGB proteins, and ORF5 (nt 5355–6275) encodes the putative CPG. CDD analysis of ORF1 revealed five putative conserved domains: Alphaflexiviridae_RdRp (accession; cd23246); Vmethyltransf (accession; pfam01660); RdRp_2 (accession; pfam00978); viral_helicase1 (accession; pfam01443), and SF1_C_RecD (accession; cd18809) in the RdRp protein. Like other potexviruses, the three conserved aa motifs HQQAKDE (aa 916–922), QDGAML (aa 1105–1110), and TFDANTE (aa 1155–1161) and a 36 aa long core motif TGX3TX3NTX22GDD (aa 1150–1185) were identified in the C-terminal region of the 1358 aa long RdRp [51,52]. To determine whether the newly identified potexvirus is a novel species or not, the HiVX genome was subjected to BLASTn and compared with the nearest alphaflexivirus sequences available in GenBank. The nucleotide sequence of the RdRp and CPG of HiVX shared 65.06–65.80%, and 66.01–69.59% nt identities with 42–55% and 18–32% query coverage with yucca alphaflexivirus 1 (MW328742.1) and agave potexvirus 1 (MW328740.1), respectively (Appendix A). The BLASTn analysis of HiVX TGB1 and TGB3 did not show any significant nucleotide identity with available virus sequences in GenBank, but TGB2 shared 73.68–77.94% nt identity with 17–24% genome coverage with the lily symptomless carlavirus (LC720794.1 and OM311155.1).

The amino acid sequence of each ORF of the HiVX genome was compared with sequences available in the public database, using BLASTp. The RdRp protein of HiVX shared 48.20–58.51% aa identity with 90–99% genome coverage with seven members [yucca alphaflexivirus 1, agave potexvirus 1, physalis virus X (PhyVX), yam virus X (YVX), Sichuan alphaflexivirus 1, clover yellow mosaic virus (ClYMV), and SMYEV] of the genus Potexvirus and/or the family Alphaflexiviridae (Figure 7A and Appendix A). The HiVX TGB1 (221 aa) and TGB2 (130 aa) proteins shared 34.21–50.45% and 41.22–53.23% aa identities with the above mentioned seven alphaflexiviruses with 97–100% and 70–100% query coverage, respectively (Appendix A). The length (64 aa) of the HiVX TGB3 is the same size as that identified for the HiCaV-C and HiCaV-D carlaviruses. The HiVX CPG and TGB3 protein shared 52.61–54.50% and 38.18–47.37% nt identities with 66–69% and 85–90% query coverage, respectively, with the four closest virus sequences [yucca alphaflexivirus 1 (QQG34624.1), PhyVX (UEP18565.1), agave potexvirus 1 (QQG34617.1) and YVX (YP_009091818.1)] available in GenBank. Even though the maximum 81% query coverage of the CPG was obtained with that of ClYMV (UZP17136) the aa identity falls from 54.50 to 32.94% (Figure 7B and Appendix A). Our individual ORF analysis data strongly supports the identification of HiVX as a new species of the genus *Potexvirus*, as all pairwise aa identities were <60%, well below the 80% aa identity species threshold (Appendix A and Figure 7A,B) [53]. The HiVX genome sequence was deposited in GenBank under accession number PP115950.

Maximum likelihood phylogenetic trees based on full genome nucleotide sequences of 56 alphaflexiviruses showed that HiVX clustered with potexviruses like PhyVX, SMYEV, and incompletely classified alphaflexiviruses including Sichuan alphaflexivirus 1, yucca alphaflexivirus 1, and agave potexvirus 1, in a highly supported group (Appendix A). To investigate the phylogenetic relationship between HiVX and the closely related viruses of the families Alphaflexiviridae and Betaflexiviridae, five more phylogenetic trees were constructed utilizing the individual ORF (RdRp, TGB1, TGB2, TGB3, and CPG) amino acid sequences using MAFFT (Figure 7C,D and Appendix A). The phylogenetic relationship based on RdRp (Figure 7C) and TGB1 (Appendix A) revealed that HiVX clusters were most closely related to Sichuan alphaflexivirus 1 followed by PhyVX, rather than yucca alphaflexivirus 1 and agave potexvirus 1, whereas PhyVX is phylogenetically more closely related to HiVX TGB3 aa sequences (Appendix A). The same consistent relationship with the pairwise comparison results was established when CPG sequences were phylogenetically analyzed (Figure 7D). An inconsistent relationship with pairwise comparison was observed when a phylogenetic tree was displayed using TGB2 aa sequences (Appendix A). In this case, HiVX clustered with the carlavirus species blueberry scorch virus (AKA93817), lily latent virus (CAB57960), and lily symptomless virus (BDP28901) rather than the PhyVX, yucca alphaflexivirus 1, agave potexvirus 1, and Sichuan alphaflexivirus 1, displayed in the phylogenetic tree (Appendix A).

### 3.7. First Confirmation of Nepovirus Infection in Hibiscus and Virus Characterization

A total of 418,388 single-end Illumina reads recovered from the hibiscus RsTHr isolate cDNA library were mapped to the PhyVNV, the second-highest proportion of viral reads (19.88%) in the sample after the betacarmovirus-related reads (Table 2). Two major contigs of 7125 and 3901 nt covering almost the entire bipartite nepovirus genome were detected in the cDNA library, and nucleotide sequences of the RNA1 and RNA2 genome segments shared 92% and 94–95% nt identities (query coverage 98–99%) with the equivalent sequences of PhyVNV (OM897214-15 and MZ357181-82) infecting cape gooseberry, in Antioquia, Colombia, and were therefore named PhyVNV-Hib. Complete sequences of the 5′- and 3′-UTRs of each genome segment of PhyVNV-Hib were not determined. The assembled contigs of PhyVNV-Hib revealed a maximum of 134 and 44 nt at the 5′ UTR and 31 and 503 nt at the 3′ end of the RNA 1 and RNA2 segments, respectively. Both 5′ and 3′ UTRs of the RNA2 segment shared 97–98% nt identities with the corresponding sequences of the PhyVNV cape gooseberry isolates BPP1 and BPP22 (MZ357182 and OM897215). Even though a partial (31 nt) 3′UTR sequence of the RNA1 segment shares 97–100% nt identity with the above-mentioned two isolates of PhyVNV, the 5′ UTR shared only 80% nt identity (query coverage 36–48%). The 3′UTRs of RNA1 and RNA2 (503–507 nt long) of both the prior PhyVNV isolates shared 98–99% sequence identities between them, as recorded for other nepoviruses like aeonium ringspot virus [54] and tomato ringspot virus [55]. The PhyVNV-Hib RNA1 and RNA2 genome sequences were deposited in GenBank under accession numbers PP115951 and PP115952, respectively.

Each RNA segment of nepoviruses codes for a single large polyprotein that is cleaved by a viral protease [56]. The PhyVNV-Hib RNA1 encodes a 2319 aa long polyprotein between nucleotide positions 135–7094, containing an RdRp domain at the nt position 1548–2030 (2.4 × 10^−45^, pfam00680) and an RNA helicase domain at nt 826–928 (2.24 × 10^−35^, pfam00910). The calculated ORF size as well as the molecular weight (MW) of the PhyVNV-Hib RNA1 polyprotein is comparatively shorter in length (2319 instead of 2324 or 2356 aa) and lower in MW (257.44 instead of 258.02 or 261.08 kDa) than those of the isolates BPP1 and BPP22. The PhyVNV isolates BPP1 and BPP22 shared 96.30% and 94.61% aa sequence identity with 100% query coverage to the PhyVNV-Hib aa sequence. Interestingly, alignment of the aa sequence of PhyVNV-Hib showed either 5 or 39 aa gaps at the N terminal region of the polyprotein between amino acids 60–126 and 301–306 when compared to the isolates BPP22 and BPP1, respectively. A similar 39 aa gap was also observed in the PhyVNV BPP1 isolate in the same position when aligned with the BPP22 isolate. The single ORF of the RNA2 of PhyVNV-Hib extends from nt 45–3398, yielding a putative polyprotein P2 of 1117 aa and a MW of 123.7 kDa. Furthermore, significant domain hits were produced to the conserved N-terminal (614–705, 7.99 × 10^−23^, pfam03689), central- (781–945, 1.10 × 10^−47^, pfam03391), and C-terminal (953–1113, 1.10 × 10^−42^, pfam03688) amino acid sequences of nepovirus coat protein domains. MotifFinder also detects a movement protein motif (pf01107) at nucleotide position 370–432, with e-value 0.003. The P2 protein of PhyVNV-Hib also shared 97–98% aa identity with cape gooseberry isolates BPP1 and BPP22 (UEP18555.1 and WCJ12503.1).

Maximum likelihood trees based on the aa alignments of the polyprotein P1 and P2 of hibiscus-infecting nepovirus reveals PhyVNV-Hib having a very close phylogenetic relationship with PhyVNV infecting cape gooseberry. Overall, PhyVNV-Hib clustered together with the nepovirus subgroup-A members: aeonium ringspot virus, potato black ringspot, and tobacco ringspot virus (Appendix A). To our knowledge, this is the first report of PhyVNV infection in any other host besides cape gooseberry, also reported in Colombia.

### 3.8. Detection of Known Hibiscus Viruses and In Silico Identification of Unknown Virus Sequences in the RsTHr Meta-Transcriptomic Library

Virome analysis detected the highest number of single-end Illumina reads (1,271,258) mapped to HCRSV, comprising 60.42% of total virus reads found in the RsTHr library. One large contig of betacarmovirus (3932 nts) was obtained which shared 94–95% nt identity with HCRSV isolates SBO (MK27967.1) from Brazil and XM (KY933060.1) from China, respectively. Complete genome sequence alignment of the HCRSV-RsTHr isolate with SBO and XM isolates revealed that 5–11 nt are apparently missing at the 5′UTR and 7 nt at the 3′UTR. The HCRSV-RsTHr isolate genome sequence was deposited in GenBank under accession number PP115955.

We have previously reported detection of a tobamovirus (HLFPV) infecting hibiscus in Colombia [7]. Although the RsTHr library yielded only 799 tobamovirus reads, the assembled reads covered 86.76% of the entire HLFPV genome. Aligned contigs shared 99% nt sequence identity with the HLFPV-J hibiscus isolate reported in Japan (NC_025381). A total of 26 and 209 nts were found missing from the 5′- and 3′-UTRs, respectively, in the HLFPV-RsTHr genome sequence. In addition, eight more gaps (of from 56 to 154 nt) were also detected when aligned with the nucleotide sequence of HLFPV-J.

A total of 65 contigs of cilevirus were detected in the RsTHr library, comprising 42,963 reads accounting for 2.04% of the total reads matching with plant virus sequences (Table 2). Obtained cilevirus reads were further categorized within the two known species at the rates of 51.50% (CiLV-C2) and 48.50% (PFGSV) of total cilevirus reads (Table 2). The PFGSV-RNA1 sequence was complete except that 25 nt of the 5′ and 86 nt at the 3′ ends are missing, compared to the PFGSV-CYPe246 isolate from Colombia (OP564895.1). Two additional distinct contigs of 775 and 3659 nt were identified as derived from PFGSV-RNA2, which shared 99% nt identity with the same passion fruit isolate (OP564896.1). Together, the contigs covered 4434 nt (88.02%) of RNA2 compared to PFGSV-CYPe246, lacking 40 nt at the 5′UTR, 310 nt of the intergenic region in between ORF1 (p15) and ORF2 (p13), and 253 nt of the 3′ UTR. The data obtained here confirm the first detection of PFGSV in hibiscus in Colombia. After bridging the gap of RNA2 segment, both the PFGSV-RNA1 and -RNA2 genome sequences were deposited in GenBank under accession numbers PP115953 and PP115954, respectively.

Initial sequence analysis identified CiLV-C2, while RT-PCR assays amplified only the hibiscus strain of CiLV-C2 (CiLV-C2H), but no citrus strain-specific CiLV-C2 amplicon was amplified. From the 22,310 CiLV-C2 reads initially identified in the RsTHr library, seven and five contigs representing CiLV-C2 RNA1 and RNA2 were assembled, respectively, and BLASTn analysis revealed the presence of two distinct hibiscus strains of CiLV-C2. Five of the seven RNA1 contigs shared 95–97% nt identity with CiLV-C2H infecting hibiscus (KC626783.1), citrus (OP761832.1), swinglea (OP761824.1), and passion fruit (MW413437.1), whereas the remaining two RNA1 contigs (of 5685 and 2112 nt) shared 89.63–91.05% nt sequence identity, respectively, with CiLV-C2H2, hibiscus isolate MLH (OP761834.1). Similarly, three (626, 1593 and 2024 nt) out of five RNA2 contigs related to CiLV-C2 shared 96.50–99.10% nt sequence identity with CiLV-C2H isolate VHFR infecting hibiscus (OP761839.1), whereas a single contig with a 3632 nt length shared only 88.29–88.41% nt sequence identity with CiLV-C2H infecting citrus (OP761833.1), hibiscus (OP761839.1), and swinglea (OP761825.1), suggesting the presence of a distinct hibiscus strain of CiLV-C2. The third CilV-C2 RNA2 contig of only 251 nt shared 99% nt sequence identity with the CiLV-C2 isolate L147V1 (JX000025) infecting citrus.

Bioinformatic tools resulted in assembly of three contigs (260, 982 and 1250 nt) related to the soymovirus HSV, but three primer pairs specific to HSV failed to confirm HSV infection using PCR. BLASTn analysis of 982 and 1250 nt revealed 72% nt identity with 40% coverage, and 68% nt identity and 69% coverage with the genome sequence of hibiscus soymovirus strain Oahu (Acc No. OP757659.1), respectively, whereas the 260 nt contig shared 68% nt identity with peanut chlorotic streak soymovirus (NC_001634) covering 83% of contig coverage. These results suggest that the soymovirus isolate detected in the hibiscus sample (RsTHr) is distinct from those previously reported in Colombia [8] and Hawaii [31].

A total of 17 and 596 single-end Illumina reads identified in the cDNA library of RsTHr were mapped to a blunervirus, and mycoviruses, respectively, which are the lowest (8.08 × 10^−4^) and next lowest (0.028%) proportions of virus reads in the library (Table 2). No primers for detection of the blunervirus, mitovirid, and totivirus genomes were designed to validate their presence in the library as only 452 nt (202 and 250 nt) blunervirus, 3575 nt mitovirid, and 409 nt (195 and 214 nt) totivirus sequences were retrieved, respectively. BLASTx analysis of 202 and 250 nt of blunervirus contigs shared 42% and 40% aa identities with 97% and 98% query coverage, respectively, with the movement protein aa sequence of tea plant necrotic ring blotch blunervirus (UXP70274.1).

Eleven mycovirus contigs (191–865 nt) were further distributed within the family *Mitoviridae* (215–865 nt) or genera *Totivirus* (195 and 214 nt), *Polymycovirus* (191 nt), and *Botybirnavirus* (252 nt). The majority of the mycoviruses detected here have either positive sense single-stranded RNA (+ssRNA) (e.g., mitovirid) or double-stranded RNA (dsRNA) (e.g., totivirus and polymycovirus) genomes. BLASTx of all seven mitovirid contigs shared 47–67% aa identity with 45–99% query coverage with the RdRp gene of Beta vulgaris mitovirus 1 (a contig of 865 nt, Acc. DAB41759.1), Xinjiang mito-like virus 73 (420, 567, and 571 nt, UPW42074.1), Paris mitovirus 1 (700 nt, QVU28731.1), Sanya mito-like virus 3 (237 nt, UUW21467.1), and Sichuan mountain mitovirus11 (215 nt, UPW42216.1). The totivirus contig (195 and 214 nt) sequences shared 57 and 61% aa identity with 98% query coverage to the hypothetical protein (WMV64394.1) and coat protein, respectively, of Uromyces fabae virus (WMV64379.1). The 191 nt polymycovirus contig shared 95% query coverage and 66% aa identity with a hypothethical protein of the RNA4 segment of Magnaporthe oryzae polymycovirus 1 (YP_010086048.1). A single contig of 252 nt shared 67% aa identity with 97% query coverage to the Didymella theifolia botybirnavirus 1 sequence (WEM04564.1).

In conclusion, ten novel virus sequences were identified as belonging to the family *Mitoviridae*, genera *Blunervirus*, *Soymovirus*, *Totivirus*, *Polymycovirus*, and *Botybirnavirus*. In addition, a possible new hibiscus strain of CiLV-C2 was also detected in a mixed infection with the above-mentioned novel and known hibiscus viruses.

### 3.9. In Silico Detection of Identified Novel Viruses in an In-House Archived Database

During BTV surveys, several *H. rosa-sinensis* samples showing unusual as well as BTV-like symptoms were collected from the Andean, Orinoquía, and coffee growing regions. In this study, we conducted a search for new and known viruses using in silico analysis of the remaining 14 mHTS libraries prepared from the hibiscus samples collected from multiple locations in Colombia. The newly discovered carlavirus (HiCaV-A, HiCaV-B, and HiCaV-C), potexvirus (HiVX), and hibiscus-infecting nepovirus (PhyVNV-Hib), plus HCRSV, HLFPV, and cilevirus sequences, were searched in the archived database to gain a comprehensive picture of their geographical distribution in Colombia.

Of the fifteen hibiscus mHTS libraries, novel carlaviruses were detected in three more mHTS libraries in addition to the RsTHr isolate prepared from the extracted total RNA of the VHFR (S54) and THL (S91) hibiscus samples collected, respectively from the Cauca Valley and Tolima departments in the Andean region, and the MGrH1 (S120) sample from Meta in the Orinoquía region. Along with other known hibiscus viruses, seven carlavirus contigs (235, 255, 284, 396, 403, 773, and 1018 nt) were detected in the VHFR mHTS library. BLASTn analysis showed 90–98% nt identities with 100% query coverage with the genome sequence of HiCaV-A. Overall, 40.88% (3364 nt) of the genomic sequence of HiCaV-A (8229 nt) was retrieved by in silico analysis. The carlavirus contig (8204 nt) obtained from the THL mHTS library shared 88% nt identity (100% query coverage) with HiCaV-B, whereas the translated amino acid of each ORF; RdRp, TGB1, TGB2, TGB3, and CPG shared 94.32, 91.77, 100, 94.12, and 95.21% aa identity. Thus, the carlavirus isolate THL detected in this sample is presumed to be a strain of HiCaV-B and the genome sequence was deposited in GenBank under accession number PP115960.

Apart from other hibiscus virus sequences, seven distinct carlavirus contigs (379, 510, 546, 870, 962, 1268, and 2410 nt) were identified as HiCaV-B genome sequences in the MGrH1 mHTS library. Five contigs (510, 546, 870, 1268, and 2410 nt) were further assembled into a large contig of 5523 nt covering nt 1635–7156 of the HiCaV-B genome identified in the sample THL. The three final aligned contigs (962, 5523, and 379 nt) of MGrH1 shared 90.73–92.62, 86.87–89.52, and 91.82–92.61% with the HiCaV-B sequences of hibiscus samples THL and RsTHr, respectively. Phylogenetically, the HiCaV-B genome sequence of MGrH1 was more closely related to RsTHr rather than the sample THL. Overall, 83.69% of the genomic sequence of HiCaV-B was recovered from MGrH1 by in silico analysis. The HiCaV-B sequences detected from samples MGrH1 and THL therefore appear to represent two possible distinct strains of HiCaV-B. HCRSV was detected in 12 samples, whereas CiLV-C2, and its hibiscus strain, and HLFPV were each detected in 11 hibiscus samples. Except for two small contigs (248 and 212 nt) of PhyVNV-RNA2 detected in the sample QCHr (S266) collected from Quindío in the coffee growing region, which shared 92–95% with BPP1 of PhyVNV (MZ357182.1), no other newly discovered virus sequences were detected in any of the other 14 mHTS libraries tested in this study.

## 4. Discussion

The genus *Hibiscus* includes over 300 species of flowering plants, which have been used for centuries for decorative and medicinal purposes; among them is *H. rosa-sinensis,* a perennial flowering plant grown throughout the seasons. Hibiscus is not native to South America, but several hibiscus species grow well in a large range from the coast to the Andean mountains. The importance of hibiscus in Colombia has increased as the hibiscus flower became part of ‘Envol Vert’ programs (https://envol-vert.org/en/prod/hibiscus-flowers/, accessed on 8 January 2024), where the farming communities grow hibiscus flowers using ecological methods and dry them using solar driers. It offers farmers an alternative source of revenue through the sale of flowers, especially during the dry season. So far, at least 23 viruses have been reported infecting ornamental hibiscus worldwide but among them only four viruses (HCRSV, HLFPV, CiLV-C2 and HSV) were previously reported from Colombia infecting hibiscus [5,6,7,8].

Metagenomics based on HTS can be used to detect known viruses and discover novel plant viruses in nature [57]. Therefore, in this study we utilized a meta-transcriptomic approach to investigate the virome associated with black spots on hibiscus leaves (RsTHr) collected from Risaralda in Colombia. Bioinformatic analysis indicates that a rich diversity of viruses can infect *H. rosa-sinensis* simultaneously. The results revealed the presence of five known viruses, (CiLV-C2, HCRSV, HLFPV, PFGSV, and PhyVNV), three novel carlavirus and one novel potexvirus species infecting a single hibiscus plant. Some of these previously known hibiscus viruses (HCRSV, HLFPV, CiLV-C2, and HSV) in Colombia have so far been reported to have a very limited host range apart from hibiscus [7,24,49,58,59,60]. Based on bioinformatic analyses of their genomic features and phylogeny, the viruses tentatively called hibiscus carlavirus A (HiCaV-A), hibiscus carlavirus B (HiCaV-B), and hibiscus carlavirus C (HiCaV-C) are proposed to be new members of the genus *Carlavirus*, and hibiscus virus X (HiVX) is proposed to be a new member of the genus *Potexvirus*, respectively, belonging to the families *Betaflexiviridae* and *Alphaflexiviridae*, in the order *Tymovirales*.

To validate the presence of the identified viral sequences in the hibiscus (RsTHr) HTS library, newly designed as well as previously published primer pairs were utilized for RT-PCR assays (Table 1 and Appendix A). Of the 11 classified virus genera and members of the *Mitoviridae* identified through mHTS and bioinformatic analysis in hibiscus, only six were detected by the RT-PCR assays. Previously published primer pairs specific to HSV [8,31] failed to confirm the presence of an HSV sequence using PCR assay. These results suggest that the soymovirus isolate contig sequences detected in RsTHr hibiscus sample might represent a novel member of the genus soymovirus rather than contamination from other sources, as no similar sequences have been recovered from other libraries. Because only small (191–865 nt) contig sequences of the novel blunervirus, totivirus, polymycovirus, and *Mitoviridae* isolates were retrieved from the RsTHr library (Table 2), validation of these genome sequences was not confirmed. The RT-PCR amplicons and the respective original HTS contigs of each of the detected viruses shared >98% nt identity with each other demonstrating high consistency between the HTS and RT-PCR data.

BLASTp and phylogenetic analyses based on the RdRp and CPG aa sequences of the novel carlaviruses showed that HiCaVs cluster closely with the SPC6V clade. They each shared 45.12–52.83% aa sequence identity with 81–100% RdRp and CPG aa sequence coverage of SPC6V (Appendix A and Figure 5A,B), indicating that HiCaV-A, HiCaV-B, and HiCaV-C could be distinct new species of the genus *Carlavirus*. Even though the concatenated aa sequences of HiCaV-D RdRp and CPG shared 51.79% and 31.43% aa identity with SPC6V (with 73% and 91% query coverage) they shared 91.21% and 94.59% aa identity with HiCaV-C (with 78 and 48% ORF coverage), respectively (Table 4). Therefore, HiCaV-D could be considered a distinct strain of the newly discovered HiCaV-C (Figure 5A,C). Recombination analysis of all the HiCaVs including the HiCaV-B sequence, assembled from the THL cDNA library, revealed that there was no true positive recombination event detected in the genome of the HiCaVs. The HiCaV-B sequence obtained from the hibiscus isolate THL is also designated as a divergent strain of HiCaV-B but not recombinant.

During the meta-transcriptomic study, we assembled a complete genome sequence of HiVX, a new potexvirus infecting hibiscus. The genome organization of HiVX is identical to that of members of the genus *Potexvirus*, family *Alphaflexiviridae*. As with carlaviruses, a distinct potexvirus species should share <72% (nt) or <80% (aa) identity for the RdRp and CPG [50,61]. We analyzed the genome properties of each ORF at both the nucleotide and amino acid level. Individual ORF analysis revealed <65% nucleotide identity and <60% amino acid identity to the RdRp and CPG sequences available in GenBank, which is well below the species threshold identity of 80% (Appendix A). A phylogenetic analysis of the complete HiVX genome, RdRp and CPG protein of different members of the family *Alphaflexiviridae* revealed that the HiVX is most closely related to members of the genus *Potexvirus*, particularly with PhyVX and the incompletely classified Sichuan alphafexivirus, agave potexvirus 1, and yucca alphaflexivirus 1 (Figure 7C,D and Appendix A).

This article reports the genome sequence of PhyVNV in hibiscus, which is the first report of PhyVNV infection in any other host apart from its first finding on cape gooseberry in Antioquia, Colombia. BLASTn searches of the databases produced significant hits and shared 92–95% nt identity with PhyVNV isolates BPP1 and BPP22 (MZ357181-82 and OM897214-15). The genomic organization and size of PhyVNV-Hib is similar to PhyVNV cape gooseberry isolates, BPP22 and BPP1, except that the RNA1 polyprotein is 5 or 39 aa shorter than for isolates BPP1 and BPP22, respectively. Previously, PhyVNV was identified in coinfection with a potexvirus (PhyVX) in the cape gooseberry [62] and our analysis confirmed that PhyVX is distinct from HiVX, though grouped in the same clade in the phylogenetic tree (Appendix A). HLRSV, the possible nepovirus reported from hibiscus in Nigeria, sometimes produced faint chlorotic spots or remained symptomless, but no serological relationship of HLRSV with the any members of the *Nepovirus* was confirmed [32]. The lack of any available sequence information for HLRSV precludes identification of any relationship to any of the PhyVNV isolates, including PhyVNV-Hib. As Antioquia borders the departments of Caldas and Risaralda to the south, the occurrence of PhyVNV in these regions could be associated with a recent emergence or adaptation of this PhyVNV nepovirus and the evolution of the novel potexvirus (HiVX) infection in hibiscus.

The PFGSV infecting hibiscus shared 48.50% of total 2.04% cilevirus reads (Table 2). More than 99% PFGSV-RNA1 and 88% of PFGSV-RNA2 genome sequences were retrieved from the RsTHr HTS library, which shared 99% nt sequence identity with Colombian passion fruit isolate CYPe246 (OP564895). This finding confirmed the detection of PFGSV in hibiscus in Colombia for the first time outside Brazil and Paraguay [17].

A total of 57 contigs were detected for CiLV-C2 comprising 51.50% of total cilevirus reads (Table 2). Except for one short contig (251 nt) of CiLV-C2 RNA2, the remaining 56 contigs were assembled into eleven large contigs representing CiLV-C2 RNA1, and -RNA2 of hibiscus-infecting isolates. Interestingly, five of the seven RNA1 and three of the four RNA2 contigs shared 95–99% nt identity with known hibiscus strain CiLV-C2H; meanwhile, the remaining two RNA1 contigs (of 5685 and 2112 nt) shared 90–91% nt sequence identity with a distinct second hibiscus strain, CiLV-C2H2 (OP761834), and the fourth RNA2 contig (3632 nt) shared only 88% nt sequence identity with CiLV-C2H infecting citrus (OP761833), hibiscus (OP761839), and swinglea (OP761825). The bioinformatic analysis confirmed the presence of two different hibiscus strain sequences of CiLV-C2 in the RsTHr HTS library. Since the genus *Cilevirus* does not have established guidelines for strain demarcation within the species, in the previous study CiLV-C2H and CiLV-C2H2 strains were determined based on <90% nt sequence identity among all available CiLV-C2 isolate sequences [7,16]. In the current study, three of the cilevirus contigs shared 90.35% nt identity with 89% RNA1 genome segment coverage to CiLV-C2H2 (OP761834.1) and 88.31% nt identity with 74.5% RNA2 genome segment coverage with CiLV-C2H infecting hibiscus (OP761839.1), respectively. Therefore, the partial RNA1 and RNA2 combination is to be considered as the presence of a third hibiscus strain of CiLV-C2, which we designated CiLV-C2H3, or a recombinant. Further research is needed to validate this distinction and to examine possible recombination or reassortment between CiLV-C2 isolates.

Even though three contigs (260, 982 and 1250 nt) related to the soymovirus HSV were retrieved from the RsTHr library, the HSV specific primer pairs failed to detect HSV using PCR assay. On the other hand, all the contigs shared 68–72% nt identity with 40–83% query coverage to HSV (OP757659) and peanut chlorotic streak soymovirus (NC_001634). These results indicate that sequence of a possible new member of the genus *Soymovirus* might be present in the RsTHr Hibiscus sample. The lowest number of virus-related single-end Illumina reads (only 17) identified in the cDNA library of RsTHr were mapped to blunervirus. Two small contigs (200 and 252 nt) related to blunervirus (family *Kitaviridae*) may represent either the first finding of this genus in hibiscus or crosstalk contamination during the run with the HTS libraries of another host. To confirm this finding more hibiscus samples from the same location need to be analyzed by HTS.

In addition to the plant virus sequences, at least 11 mycovirus-related (mitovirus, family *Mitoviridae*; and totivirus, polymycovirus, and botybirnavirus) contigs were also detected in the RsTHr HTS library (Table 2). Many of the obtained contigs had only short genomic mycovirus-like sequences (191–865 nt). For future validation of the occurrence of mycovirus-like sequences associated with hibiscus, the leaf meta-RNA-seq analyses should be undertaken to determine the entire genome of the mycoviral communities in hibiscus-associated fungal populations.

Three new carlaviruses (HiCaV-A, HiCaV-B, and HiCaV-C) and one possible distinct strain of HiCaV-C (HiCaV-D) were identified for the first time in *H. rosa-sinensis.* Even though there was a finding of a carlavirus sequence in an *H. rosa-sinensis* sample from Guangzhou province in China [33], which shared 79.6% nt identity with potato virus M (MT114149), no correlation was established between the presence of the Chinese hibiscus carlavirus sequence and the observed leaf rolling, deformation, or chlorosis symptoms in *H. rosa-sinensis*. Moreover, no aligned PVM-like sequence or raw HTS data are available in GenBank for the sequence comparison. Carlavirus reads accounted for the third highest proportion (10.24%) after betacarmovirus (60.42%) and nepovirus (19.88%) related reads in the RsTHr library (Table 2), suggesting that symptoms on hibiscus might be caused by multiple viruses in synergistic interaction. Earlier studies have shown that the CRP encoded by ORF6 in carlaviruses plays a critical role in determining symptoms and pathogenicity [63,64] but, unfortunately, none of the new carlaviruses were shown to have this protein coding ORF. Furthermore, bioinformatic analysis from all the four HiCaVs HTS libraries confirmed that there was no contig similar to the CRP presence. Interestingly, the RdRp and CP of HiCaV-A, HiCaV-B, and HiCaV-C were each shown to have relatively close relationships to those of SPC6V (Appendix A), falling within the same clade in their phylogenetic trees (Figure 5C,D). SPC6V was first found in a sweet potato cultivar from the Dominican Republic, in geographic proximity to Colombia, and presumably sharing a common ancestor. SPC6V, like the HiCaVs, lacks an identifiable ORF6 CRP downstream of the CP; instead, SPC6V encodes a protein lacking similarity to any other known proteins [65]. This analysis was reconfirmed by BLASTp analysis even 10 years after the discovery of SPC6V. It is possible that reads derived from an atypical ORF6 of the HiCaVs were present but not identified as being of viral origin due to a similar lack of identity to known viral sequences. In the absence of a detectable ORF6 associated with pathogenicity, the new carlaviruses may not be associated with the black spot symptoms unique to hibiscus sample RsTHr.

In the RsTHr library, the fourth highest proportion of virus-related reads was detected for the potexvirus HiVX (7.29%). Natural infection associated with potexviruses in different host plants may show mosaic, mottle, necrosis, chlorosis, spots, or dwarf symptoms, or may be symptomless [66]. However, *Cattleya* and *Cymbidium* orchid leaves and flowers infected by cymbidium mosaic virus can show dark necrotic spots and streaks [67], and the waxy cuticle of hibiscus leaves might well make similar necrosis appear black. Notably, HiVX has only been detected in sample RsTHr, the only sample in which the black spots have been observed, suggesting the possibility that HiVX infection induces the black spot symptom. However, considering the large number of distinct viruses present in sample RsTHr, and the presence of HCRSV and cileviruses in the mixed infection, which are each known to cause chlorotic rings or spots, the effects of a particular combination of viruses may be required to induce the observed black spots. HCRSV induces leaf mottling and/or chlorotic rings or spots symptoms and overall HLFPV infection also produce chlorotic mottle symptoms [24], whereas cileviruses induce green ringspots with internal chlorotic spots in senescing leaves or irregular green rings, necrotic ringspots, or chlorotic round lesions with green borders [7,16,17]. Several viruses may cause similar symptoms but the symptom expression in infected plants could be different with viral load, presence of virus variants, or virus combinations in a mixed infection as seen in papaya virus infection [68]. The biological significance of symptoms association with the mixed virus population is worth investigating in the future. However, unless HiVX and the other viruses can be separated and recombined in healthy plants by artificial inoculation, or less complex natural mixed infections are identified in plants displaying black spots, it may not be possible to associate the black spot symptom with a particular virus or mixed infection.

Out of 30 symptomatic hibiscus leaf samples collected from multiple regions in Colombia, a total of 15 were sequenced using HTS and an in-house database was created to study hibiscus virome. The VHFR and THL are two of the six samples collected from the Cauca Valley and Tolima departments, respectively, and one (MGrH1) of the seven samples collected from Meta were each identified to be positive with one of the newly discovered carlaviruses. The carlaviruses detected in THL and MGrH1 samples are presumed to be two distinct strains of HiCaV-B. Overall, the novel carlavirus sequence HiCaV-A was detected in hibiscus sample VHFR from Cauca Valley in Andean region and HiCaV-B sequence was detected both in the Andean (the THL sample from Tolima) and in the Orinoquía regions (the MGrH1 sample from Meta). Except for the RsTHr sample from Risaralda, no HiCaV-C-like sequence was retrieved from the in-house archived hibiscus database. Of the 14 samples, only two small contigs of PhyVNV-RNA2 were detected in the sample QCHr collected from Quindío in the coffee growing region. The newly discovered potexvirus sequence was not detected in any of these HTS libraries. In contrast PFGSV was detected only in the two samples collected from Quindío and Risaralda in the coffee growing regions. It would be interesting to completement these in silico data with the RT-PCR followed by Sanger sequencing analysis, but due to the unavailability of the original samples or archived total RNA, we did not conduct these assays.

From the epidemiological point of view, it will be worthwhile in future to study (i) the potential sources of infection, (ii) the transmission pathways of the newly discovered viruses, (iii) the causal agent/s associated with black spot disease of hibiscus, and (iv) the potential threat to the surrounding plant species or major crops growing in Colombia.

## 5. Conclusions

In this study we report the genome sequences of three new carlaviruses, including HiCaV-A, HiCaV-B, HiCaV-C, and the new potexvirus HiVX, infecting *H. rosa-sinensis* in Colombia. In addition, the RNA-seq data analyses suggested the presence of distinct HiCaV isolates belonging to newly discovered carlavirus species HiCaV-A and HiCaV-B in both the studied sample and archived database, while HiCaV-C and HiCaV-D, (a distinct variant of HiCaV-C), were detected only in the studied sample RsTHr. This manuscript also reports the first incidence of PhyVNV infection in hibiscus, in addition to its original host cape gooseberry. Furthermore, the meta-transcriptomic approach identified PFGSV for the first-time in hibiscus in Colombia and revealed the possibility of a third hibiscus strain of CiLV-C2 (CiLV-C2H3) present in nature. The findings of this study will provide useful information for the development of rapid, sensitive, and reliable molecular tools to prevent the introduction of these newly described viruses to an uninvaded geographic location.

## Figures and Tables

**Figure 1 viruses-16-00267-f001:**
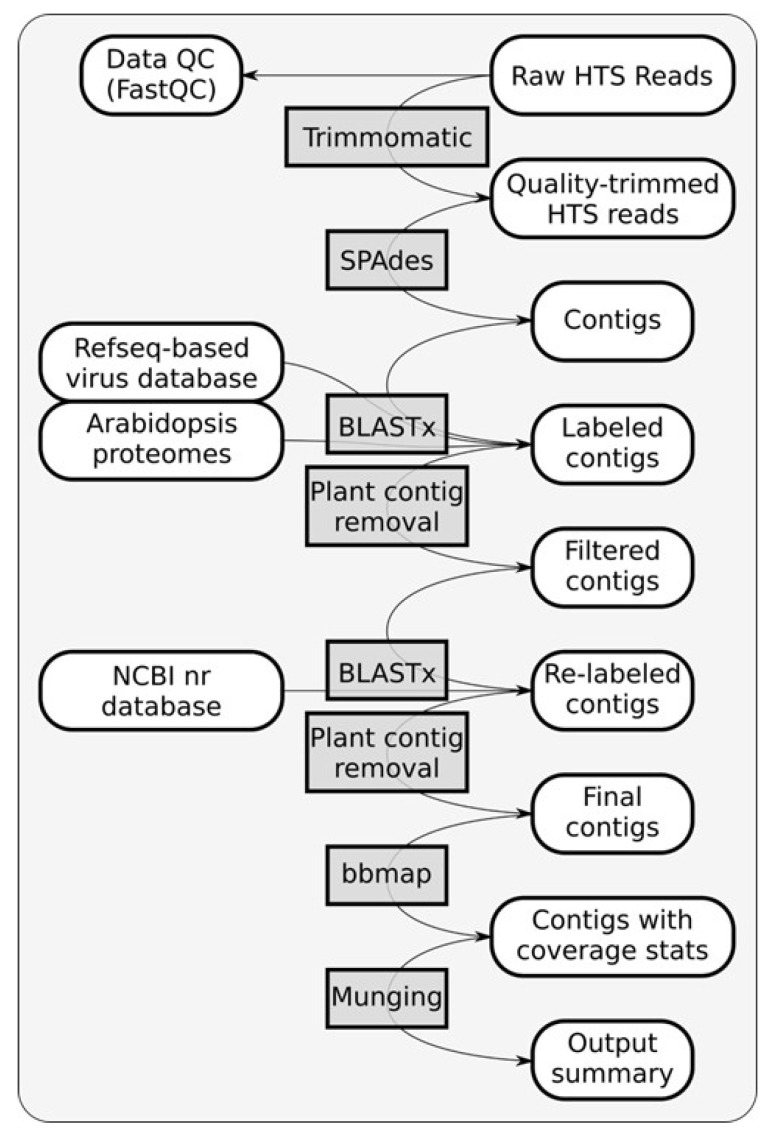
Bioinformatic pipeline utilized for plant virus detection and discovery of new viruses from infected hibiscus plants.

**Figure 2 viruses-16-00267-f002:**
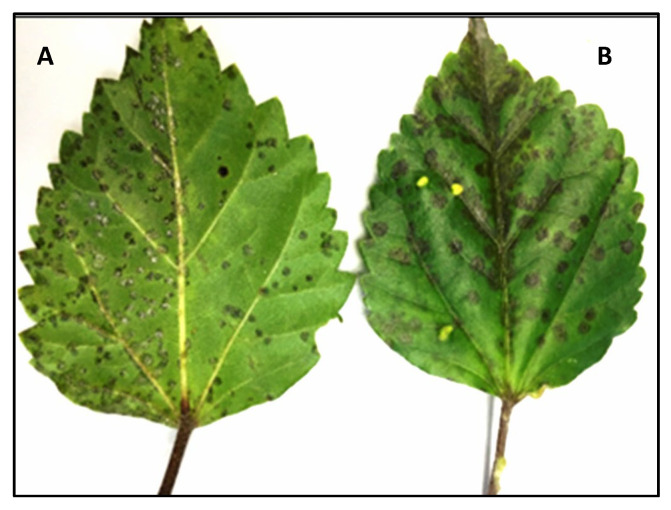
Black spot symptom observation on (**A**) ventral or abaxial and (**B**) dorsal or adaxial sites in *Hibiscus rosa-sinesis* leaves. Viruses detected in the single hibiscus leaf sample are listed in Table 2.

**Figure 3 viruses-16-00267-f003:**
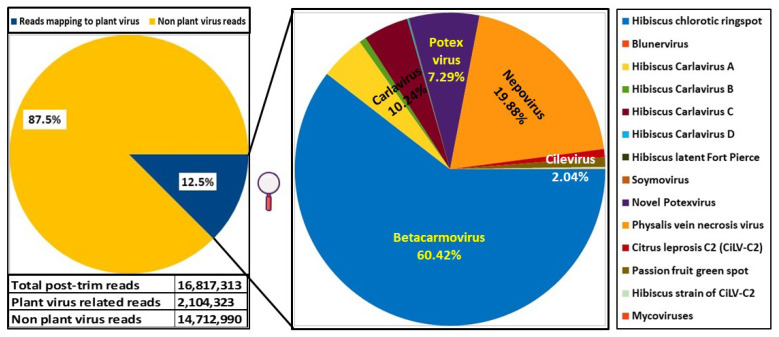
Reads obtained from the Ribo-Zero cDNA library of the RsTHr hibiscus sample showing the total raw reads and the distribution of genus- and species-specific virus reads in the pie charts. NB: Colors representing viruses, like the hibiscus latent Fort Pierce virus, mycoviruses, soymovirus and blunervirus, are not visible in the pie chart as their reads varied from only 17 to 799 nt (collectively representing only 0.13% of the virus-specific reads).

**Figure 4 viruses-16-00267-f004:**
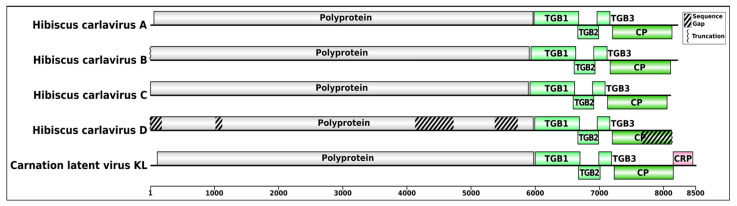
Schematic genomic organization of four newly discovered carlaviruses (hibiscus carlavirus A, hibiscus carlavirus B, hibiscus carlavirus C, and hibiscus carlavirus D) with the *Carlavirus* type species carnation latent virus. Gap between ORFs polyprotein/TGB1 and between TGB3/CP are the intergenic regions. The cross-hatched regions in the hibiscus carlavirus D genome are the portions of the genome missing from the HTS data.

**Figure 5 viruses-16-00267-f005:**
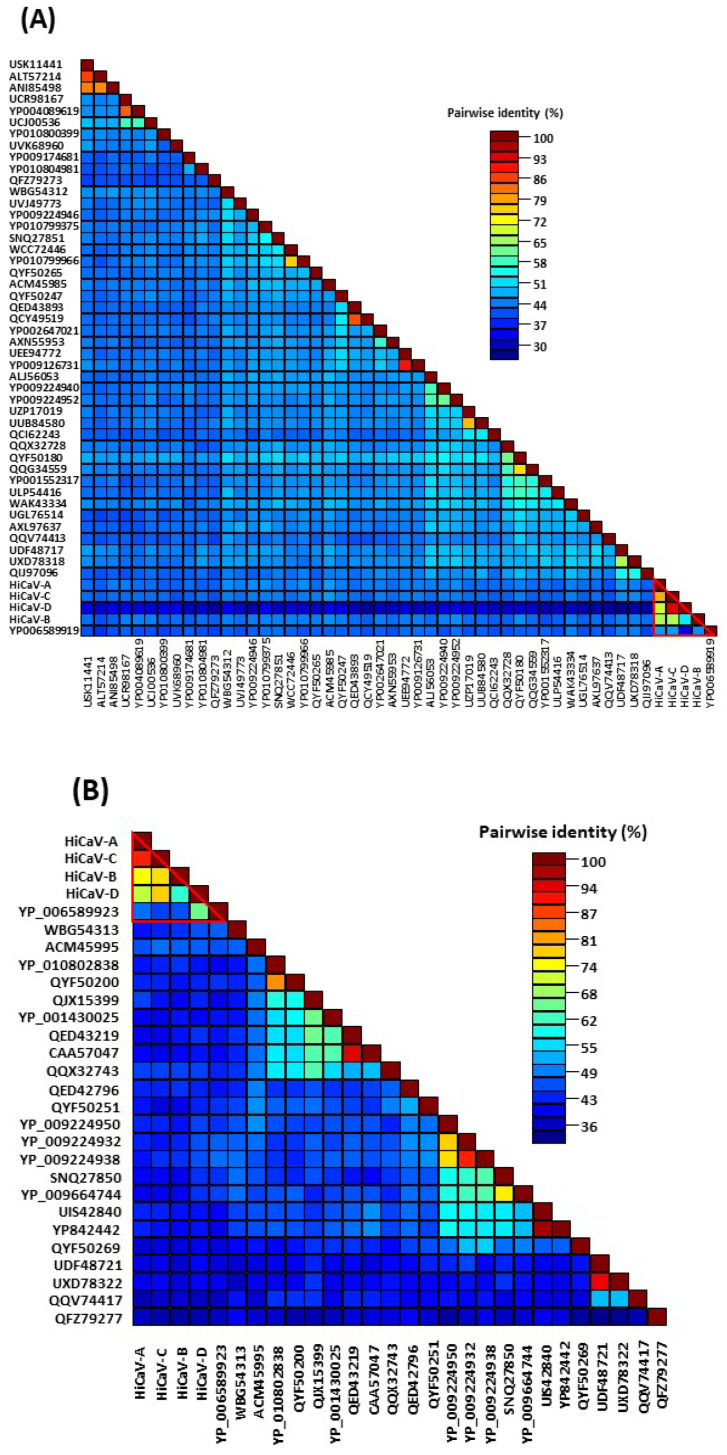
Heatmap analysis of the amino acid pairwise identity matrixes of hibiscus carlaviruses (HiCaV-A, HiCaV-B, HiCaV-C, and HiCaV-D) infecting hibiscus sample RsTHr. In total, 24 to 46 carlavirus species from the NCBI database were included at whole replicase-related protein (RdRp) level (**A**) and coat protein (CP) level (**B**), respectively. The square boxes inside the red-outlined triangles in Figure 5A,B represent the maximum 45–53% and 47–52% pairwise identity of the RdRp and CPG of the HiCaVs (HiCaV-A, HiCaV-B, HiCaV-C, and concatenated HiCaV-D) matrixes, respectively, with sweet potato C6 virus (YP_006589919). Phylogenetic relationships of the hibiscus carlaviruses (HiCaV-A, HiCaV-B, HiCaV-C, and HiCaV-D; in red box) identified in the RsTHr high-throughput sequencing library with 46 complete RNA-dependent RNA polymerase (**C**) and 24 coat protein (**D**) gene sequences of carlavirus species using the Multiple Alignment using Fast Fourier Transform (MAFFT, v7.520) program with bootstrap values of 1000 replicates. The accession numbers represent the viruses used in the heatmap analysis and their names corresponding to the accession numbers are displayed in the phylogenetic trees created using the RdRp (**C**) and CP (**D**) amino acid sequences.

**Figure 6 viruses-16-00267-f006:**
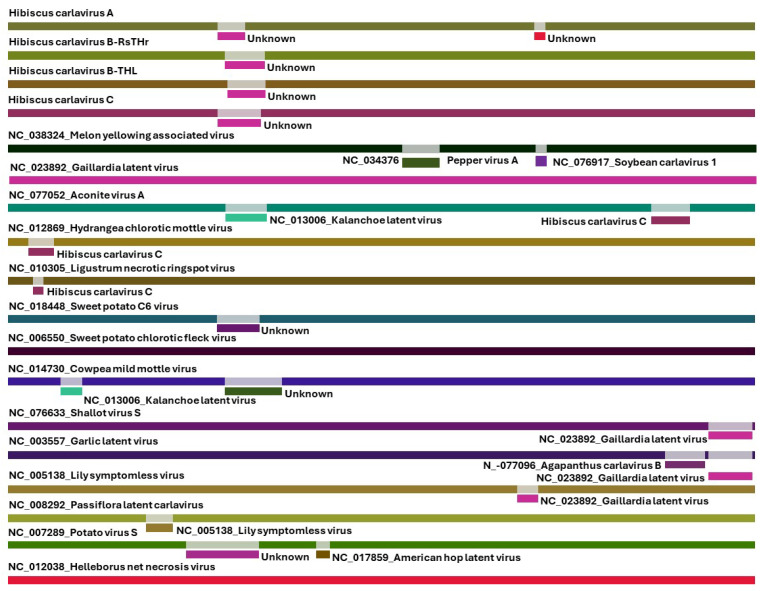
Recombination analysis by RDP4 (v4.101) identified positions of unique recombination events in the hibiscus carlaviruses (HiCaV-A, HiCaV-B-RsTHr, HiCaV-B-THL, and HiCaV-C)-aligned sequences in relation to the other 64 complete carlavirus species sequences available in GenBank. The current figure included only 14 closely related carlavirus species. If the source of the minor parent sequence was confirmed by RDP analysis, then the name was given to the side of each color, otherwise the possible minor parent’s name was written as unknown.

**Figure 7 viruses-16-00267-f007:**
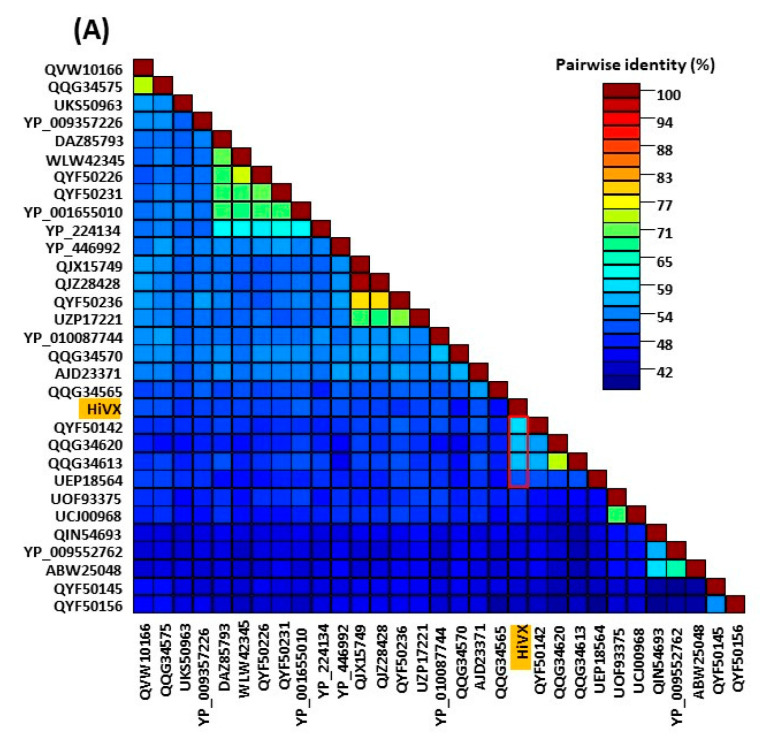
Heatmap analysis of the pairwise amino acid identity matrixes of hibiscus potexvirus (HiVX) infecting hibiscus sample RsTHr with 30 RdRp (**A**) and 26 coat protein (**B**) potexvirus sequences from the NCBI database. The red-outlined rectangular box in (**A**) represents the maximum pairwise identity (54–58%) matrixes of HiVX-RdRp with Sichuan alphaflexivirus 1 (QYF50142), yucca alphaflexivirus 1 (QQG34620), and agave potex virus 1 (QQG34613), followed by 49% identity with physalis virus X (UEP18564). The red-outlined rectangular box in (**B**) represents the maximum pairwise identity (53–55%) matrixes of HiVX-RdRp with yucca alphaflexivirus 1 (QQG34624) and agave potex virus 1 (QQG34617), followed by 35% identity with physalis virus X (UEP18568). Phylogenetic relationships of the hibiscus potexvirus (HiVX) identified in the RsTHr high-throughput sequencing library with 30 complete RNA-dependent RNA polymerase (**C**) and 26 coat protein (**D**) gene sequences of the potexvirus species using the Multiple Alignment using Fast Fourier Transform (MAFFT, v7.520) program with bootstrap values of 1000 replicates. The accession numbers represent the viruses used in the heatmap analysis and their names corresponding to the accession numbers are displayed in the phylogenetic trees created using the RdRp (**C**) and CP (**D**) amino acid sequences. Newly discovered HiVX virus position in the heat map (**A**,**B**) and phylogenetic trees (**C**,**D**) are highlighted in yellow color.

**Table 1 viruses-16-00267-t001:** Virus species-specific primer sequences used for reverse transcription–polymerase chain reaction assays and their polarity, length, melting temperature, GC percentage, amplified region, and amplicon size for each primer pair and its nucleotide and amino acid identity with nearest relatives are detailed.

Virus Abbreviation (Genus)	Polarity	Primer		Length (nt)	GC Content (%)	Melting Temperature (°C)	Amplified Region	Amplicon Size (nts)	Blast Analysis of the RT-PCR Amplicons					
		Name	Sequence from 5′ to 3′						Query Coverage	Nucleotide Identity	Virus Hit (Accession No.)	Query Coverage	Amino Acid Identity	Virus Hit (Accession No.)
HiCaV-A (*Carlavirus*)	sense	HiCaV-A-1F	AAG ATT TAT GCC CCA ATG CCA TTG TT	26	38.5	57.9	RdRp	434	79	65	Apple stem pitting virus (MZ147976.1)	98	59	Elderberry carlavirus E (YP_009224952.1)
	antisense	HiCaV-A-1R	TGA AAC AGC ATC AAA TGG GCC AAA	24	41.7	58.2								
	sense	HiCaV-A-2F	CAG GCG ATC CAT GTC AAA GTG ATT	24	45.8	57.6		707	16	72	Asian prunus virus 3 (KR998050.1)	99	53	Ligustrum virus A (YP_009288956.1)
	antisense	HiCaV-A-2R	AAA TTC TTC AAT GTC CTC TTG CTG CC	26	42.3	58								
HiCaV-B (*Carlavirus*)	sense	HiCaV-B-1F	AAG TTA AGC ACC TTA GGG ATA GGA GC	26	46.2	58	RdRp	761	No identity found			No identity found		
	antisense	HiCaV-B-1R	GCT TCA ATR GAG GAG TGC AAT CG	23	50	57.9								
	sense	HiCaV-B-2F	TGA CGA GTA TCA GTT AGG CTA CAG TGA	27	44.4	58.4	TGB1 and 2	702	29	69	Garlic common latent virus (MN059118.1)	63/38	44/59	Chrysanthemum virus B (CAE54958.2)/Sweet potato C6 virus (YP_006589919.1)
	antisense	HiCaV-B-2R	AGA AGA TGC ACA AGA TAG TGG TGA AGA G	28	42.9	58.2								
HiCaV-C (*Carlavirus*)	sense	HiCaV-C-1F	GAG GCT ATC TCC TCT CCA CAA GTA AG	26	50	57.8	RdRp	711	36	67	Cowpea mild mottle virus (Acc. No. MW201798.1)	94/98	49/41	Sweet potato C6 virus (YP_006589919.1)/Cowpea mild mottle virus (QTF19713.1 and UOF93340.1)
	antisense	HiCaV-C-1R	CAA CCA CAA GCT TTG GAA TGA GGA G	25	48	58.3								
	sense	HiCaV-C-2F	AGA AGC CAC AAT TGG TCT TTG AAA GAA TG	29	37.9	58.1	RdRp and TGB1	675	33	72		32/33	77/72	
	antisense	HiCaV-C-2R	TGG AAT CAA TGC TGA AGC CAA TTG AG	26	42.3	58.1								
HiCaV-D (*Carlavirus*)	sense	HiCaV-D-1F	TGG AGG TTA CCT CCT TTC TAC CAG TA	26	46.2	58.3	RdRp	696	15	100	Sweet potato C6 virus (Acc. No. NC_018448.1)	97/89	37/46	Cowpea mild mottle virus (AUM57217.1)/Sweet potato C6 virus (YP_006589919.1)
	antisense	HiCaV-D-1R	ATA ACA AGA ATG TCA CAG AAG CTG CAT GA	29	37.9	58.6								
	sense	HiCaV-D-2F	AGT TGA CCA TTC ACA TGA AGT TAG AGA TGG	30	40	58.3		464	8	100		99/95	35/37	Helleborus net necrosis virus (YP_002574614.1)/Rose virus A (YP_010799966.1)
	antisense	HiCaV-D-3R	AGC CAG AGC TCT AGC TCT CTC TAT AC	26	50	58.4								
HiVX (*Potexvirus*)	sense	HiVX-RdRp-F	GTT CAA AGG TAA GGA GAG ACA GTA CGA C	28	46.4	58.2	RdRp	724	31/30	68/72	Papaya mosaic virus (NC_001748.1)/Plantago asiatica mosaic virus (KU697313.1)	99	51	Cardamom virus X (DAZ85788.1) and Agave potexvirus 1 (QQG34613.1)
	antisense	HiVX-RdRp-R	GTT GGC TAC GAA CTT GAC GTT TCT G	25	48	58.2								
	sense	HiVX-CPG-F	CCT TCA GAC GCT GAA ATT GAA GCT G	25	48	58.4	CPG	701	42/23	66/70	Yucca alphaflexivirus 1 (MW328742.1)/Agave potexvirus 1 (MW328740.1)	59	54–55	Yucca alphaflexivirus 1 (QQG34624.1) and Agave potexvirus 1 (QQG34617.1)
	antisense	HiVX-CPG-R	AGG ACT CCC TGA AAG AAG TCG AAA G	25	48	58.2								
PhyVNV-Hib (*Nepovirus*)	sense	HiNeV-RNA1-F	ATG AGG TCG TTA TGA AGG GTA AGC A	25	44	57.8	Polyprotein-1	828	100	93	Physalis vein necrosis virus (OM897214.1 and OM897215.1)	99	98	Physalis vein necrosis virus (UEP18554.1 and UEP18555.1)
	antisense	HiNeV-RNA1-R	CAA CAT CAT CGT CAA CAT CTT CAA ATG AGG	30	40	58.1								
	sense	HiNeV-RNA2-F	ACG TGA AAA ATC TAA GGT GCT GTC TCT	27	40.7	58.2	Polyprotein-2	651	100	94		99	97	
	antisense	HiNeV-RNA2-R	TCT GGT ATC TTG GCA GTA TGG AGA TG	26	46.2	57.8								
HCRSV (*Betacarmovirus*)	sense	HCRSV-CPG-F	AGC TGT CAA CAC TTA AGG TTA CAG CT	26	42.3	58.2	CPG	765	100	97	Hibiscus chlorotic ringspot virus (MK279671.1)	99	99	Hibiscus chlorotic ringspot virus (AZL87708.1)
	antisense	HCRSV-CPG-R	TTG AAC TCT TCT GCT TCA CCT CCA	24	45.8	58.1								
HLFPV (*Tobamovirus*)	sense	HLFPV-CPG-F	TGT CTT ACT CGA ACA TAA CAG CTC TTA ACC	30	40	58	CPG	442	100	99	Hibiscus latent Fort Pierce virus (MT942636.1)	99	100	Hibiscus latent Fort Pierce virus (QZL11038.1)
	antisense	HLFPV-CPG-R	GCA ATC TCG AAA GAA GCC TGA TTG TAC	27	44.4	58								

**Table 2 viruses-16-00267-t002:** The initial annotation results of the contigs assembled from High-throughput sequencing (HTS) reads generated from the hibiscus black spot symptomatic leaf sample. The initial BLASTx annotations in this table reflect the most closely related viruses present in the NCBI database, but these initial identifications are, in several cases, below the threshold for species discrimination for ICTV classification with previously characterized viruses, as further discussed in the text.

Virus Detected in RsTHr HTS Library	Genus	Sum Contig (nt)	Sum Reads	Number of Contig	Degree of Coverage (%)
Hibiscus chlorotic ringspot virus	*Betacarmovirus*	3931	1,271,258	1	97.7
Physalis vein necrosis virus	*Nepovirus*	10,508	418,388	2	96.5
Elderberry carlavirus E	*Carlavirus*	835	599	1	58.0
Elm carlavirus		1141	376	1	41.8
Garlic common latent virus		8413	98,603	2	39.7
Helleborus net necrosis virus		426	147	1	68.1
Ligustrum virus A		995	297	2	52.6
Narcissus common latent virus		8229	98,212	1	40.7
Red clover carlavirus 1		595	249	1	42.2
Shallot latent virus		412	163	1	78.8
Sweet potato C6 virus		13,289	16,728	8	46.4
Physalis virus X	*Potexvirus*	6426	153,362	1	49.3
Passion fruit green spot virus	*Cilevirus*	13,109	20,833	8	99.5
Citrus leprosis virus C2		18,157	16,562	31	95.5
Hibiscus strain of CiLV-C2		10,800	5568	26	97.8
Hibiscus latent Fort Pierce virus	*Tobamovirus*	5557	799	13	99.7
Peanut chlorotic streak virus	*Soymovirus*	2492	790	3	62.9
Tea plant necrotic ring blotch virus	*Blunervirus*	452	17	2	41.6
Beta vulgaris mitovirus 1	*Mitoviridae*	865	85	1	53.8
Xinjiang mito-like virus 73		1558	332	3	55.65
Paris mitovirus1		700	129	1	46.95
Sichuan mountain mitovirus 11		215	9	1	58.82
Sanya mito-like virus 3		237	14	1	56.25
Uromyces fabae virus	*Totivirus*	195	13	2	61.87
Didymella theifolia Botybirnavirus 1	*Botybirnavirus*	252	9	1	67.07
Magnaporthe oryzae polymycovirus 1	*Polymycovirus*	191	5	1	65.57

**Table 3 viruses-16-00267-t003:** Nucleotide position of each open reading frame and untranslated regions of hibiscus carlavirus (HiCaV)-A, HiCaV-B, HiCaV-C, and HiCaV-D genomes.

Virus Name	Open Reading Frames (ORFs) and Untranslated Regions of Hibiscus Carlaviruses (HiCaVs)																									
	5′UTR ^$^			ORF1/RdRp				TGB1				TGB2				TGB3				CPG				3′UTR ^$^		
	Start	End	Length	Start	Stop	Length		Start	Stop	Length		Start	Stop	Length		Start	Stop	Length		Start	Stop	Length		Start	End	Length
						NT	AA			NT	AA			NT	AA			NT	AA			NT	AA			
HiCaV-A	1	55	55	56	5965	5910	1969	5992	6687	696	231	6665	6988	324	107	6964	7164	201	66	7205	8131	927	308	8132	8229	98
HiCaV-B	1	46	46	47	5962	5916	1971	5990	6685	696	231	6663	6986	324	107	6968	7174	207	68	7217	8158	942	313	8159	8202	44
HiCaV-C	1	2	2	3	5891	5889	1962	5919	6614	696	231	6592	6912	321	106	6894	7088	195	64	7128	8054	927	308	8055	8117	63
HiCaV-D ^#^	UD	UD	UD	3 ^#^	5786 ^#^	5784 ^#^	1927 ^#^	5959 ^#^	6636 ^#^	678	225	6614 ^#^	6934 ^#^	321	106	6916 ^#^	7110 ^#^	195	64	7150 ^#^	8076 ^#^	927 *	308 *	8077 ^#^	UD	UD

‘#’ missing nucleotide number and its position in the HiCaV-D genome was predicted based on the HiCaV-C genome sequence. * Only partial 48% genome coverage was identified, and CPG size was predicted based on HiCaV-C ORF5. ‘$’ 3′ UTR and 5′ UTR were not determined using RACE.

**Table 4 viruses-16-00267-t004:** Percentage of nucleotides and amino acid identities with genome coverage in bracket between four novel hibiscus carlavirus sequences (HiCaV-A, HiCaV-B, HiCaV-C, and HiCaV-D) were compared based on complete genome and each open reading frame. Here, concatenated RdRp and coat protein genes along with complete TGBs of HiCaV-D were used for comparison.

Sl. No.	Virus	Full Genome				ORF1 (RdRp)				ORF5 (CPG)			
		I	II	III	IV	I	II	III	IV	I	II	III	IV
I	HiCaV-A	*	69.23 (83)	72.33 (91)	77.76 (68)	*	70.18 (77)	71.49 (89)	77.35 (64)	*	76.07 (82)	80.55 (87)	73.33 (47)
II	HiCaV-B	ND	*	68.96 (77)	69.31 (55)	66.43 (99)	*	70.26 (77)	72.11 (54)	75.66 (96)	*	79.22 (73)	76.67 (22)
III	HiCaV-C	ND	ND	*	86.40 (78)	78.36 (100)	66.87 (99)	*	86.48 (78)	89.61 (100)	79.26 (95)	*	92.57 (47)
IV	HiCaV-D	ND	ND	ND	*	74.91 (78)	60.39 (78)	91.21 (78)	*	79.73 (48)	59.57 (45)	94.59 (48)	*
		**ORF2 (TGB1)**				**ORF3 (TGB2)**				**ORF4 (TGB3)**			
I	HiCaV-A	*	66.01(100)	68.95 (100)	69.74(100)	*	73.40 (57)	72.22 (99)	74.05 (96)	*	86.67 (14)	71.73 (92)	71.35 (94)
II	HiCaV-B	64.94 (100)	*	70.75 (65)	73.40 (19)	64.49 (100)	*	74.87 (57)	74.48 (57)	47.83 (96)	*	NS	NS
III	HiCaV-C	72.73 (100)	62.77 (100)	*	84.20 (100)	85.05 (100)	65.42 (100)	*	88.47 (100)	73.44 (96)	51.16 (63)	*	90.26 (100)
IV	HiCaV-D	71.86 (100)	61.04 (100)	87.88 (100)	*	84.11 (100)	64.49 (100)	97.17 (100)	*	75 (96)	53.06 (66)	90.62 (100)	*

**Table 5 viruses-16-00267-t005:** Multiple recombination-detecting programs (RDPs) detect the recombination sites in the genome of hibiscus carlaviruses (HiCaV-A, HiCaV-B-RsTHr, HiCaV-B-THL, and HiCaV-C). All four isolates were identified as recombinant using the recombination-detecting software: R (RDP), M (MaxChi), C (Chimaera), and Si (SiScan) programs in RDP4 (v.4.101).

Recombinant Virus	Breakpoint Position (Nts)				Parent ^a^				Software	*p*-Value ^b^
	In Alignment		Without Gaps		Major Parent		Minor Parent			
	Beginning	Ending	Beginning	Ending	Name (Accession No.)	% Similarity	Name (Accession No.)	Unknown		
Hibiscus carlavirus A	3355−3798	3688−3937	2099−2310	2228−2400	Melon yellowing associated virus (NC_038324)	48	Gaillardia latent virus (NC_023892)	UK	RMCSi	3.143 × 10^−3^
	8402−8460	8496−8721	6044−6102	6138−6342	Sweet potato C6 virus (NC_018448)	53.8	Helleborus net necrosis virus (NC_012038)	UK	Rsi	1.633 × 10^−3^
Hibiscus carlavirus B-RsTHr	2465−3644	3974−4472	1798−2171	2372−2635	Melon yellowing associated virus (NC_038324)	47.7	Gaillardia latent virus (NC_023892)	UK	RMCSi	1.257 × 10^−4^
Hibiscus carlavirus B-THL	2465−3644	397−4472	1853−2226	2427−2690		47.6		UK	RMCSi	1.257 × 10^−4^
Hibiscus carlavirus C	3354−4048	3937−3688	2163−2335	2033−2396		48.2		UK	RMCSi	3.143 × 10^−3^

^a^ The identified parent sequences have a percentage of contribution in the recombination event, but a probable minor parent does not show the percentage of recombination contribution in RDP4, written in table as ‘UK’ for Unknown. ^b^ The reported *p*-value mentioned in the far right column is for the program in bold type in RDP4 for the recombination detecting software and mentions the greatest calculated *p*-value among the carlaviruses.

## Data Availability

The data presented in the study are deposited in the NCBI repository at https://www.ncbi.nlm.nih.gov/, under accession numbers PP115947-PP115960 and BioProject ID PRJNA1063669. Accession numbers were provided on 12 January 2024, and will be available immediately in the public database after publication of the manuscript.

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
