# Peer review of "Meta-Transcriptomic Analysis Uncovers the Presence of Four Novel Viruses and Multiple Known Virus Genera in a Single Hibiscus rosa-sinensis Plant in Colombia"

_viruses, 2024, doi:10.3390/v16020267_

Round 1
Reviewer 1 Report
Comments and Suggestions for Authors
Abstract should be rewritten as:
Line 19: „ Bioinformatic analysis identified 12.5% of total reads in the library which mapped to viral genomes“ this sentence in the abstract is somewhat uninformative, as the reader don´t know how the template was prepared (total RNA? depleted or not etc.), either to complement the information or remove this sentence
Lines 27-30 bring the same information as mentioned in lines 20-22
Lines 31-32: „Overall, the meta-transcriptome analysis OF? identified virome provides significant insights into the discovery and diversity of virus genera in a single hibiscus plant.“ This sentence is too vague and without a deeper meaning. Authors should rather emphasize the heterologous nature of virome or some practical aspects of their discoveries.
Line 421: figure 5A, 5B should be moved to the Supplementary data, as 5C and 5D are much more informative, the same for line 627 (7A, 7B)
Line 543: the HiCaV-B-THL is for the first time mentioned here, but no information about this sample/isolate is done in the Material and methods, it is rather confusing and hard to follow. Only in line 791, there is an information about these additional HTS datasets.
Line 838: it should be mentioned if it is a tentative species names given by authors for three novel carlaviruses (HiCaV-A, HiCaV-B and HiCaV-C), and one novel potexvirus (HiVX)
Line 997: although the partial sequences related to novel carla and potexviruses have been identified within HTS dataset, it should be interesting to complement these in silico data with the RT-PCR/sanger sequencing analysis of environmental samples from Hibiscus, allowing to obtain the partial sequence data enabling the evaluation of potential intra-species diversity. As such, the present work remain only in silico analysis of a few HTS datasets.
Overall, the manuscript is very carefully, precisely (and in detail) written, however, paradoxically it makes it intricate and very hard to follow. The reader is lost in huge amount of data and sometimes, it is difficult to understand it. A big part of the Discussion section only repeats the Results data (probably, for better clarity, I would suggest to join Results and discussion into one section.
This is not to decrease the interest of this work (containing very interesting data), however, in this case a simplicity and brevity can be more effective and powerful than complexity and abundance.
Reviewer 2 Report
Comments and Suggestions for Authors
In the paper entitled ,,Meta-transcriptomic analysis uncovers the presence of four novel virus and multiple known virus genera in a single Hibiscus rosa-sinensis plant in Colombia” submitted to ,,Viruses” under ID viruses-2848632 the Authors described the results of high-throughput sequencing of one sample originated from hibiscus displaying black spots on upper and lower sides of most of the leaves. The presence of ten viruses were identified, which includes five known hibiscus infecting viruses (CiLV- C2, CiLV-C2H, HCRSV, HLFPV, and PFGSV), physalis vein necrosis virus (PhyVNV), three novel carlaviruses, and one novel potexvirus. Metagenomics approaches have a huge potential for detecting viruses, characterising viromes and exploring virus emergence and epidemiology. The paper is interesting and described detection of many different virus species in one single plant sample. Nevertheless, some points need to be clarified and my recommendation is minor revision. Please find my specific comments below:
Line 89-94, page 2 - this part describes the results and should be deleted from Introduction
Line 131, page 3 – at instead of @
Line 149, page 3 – Assembled contigs were searched against an in-house database comprising of filtered viral protein sequences from RefSeq and the proteomes of Arabidopsis thaliana and A. lyrata (Uniprot taxon IDs 3702 and 81972). It is not clear why proteomes of Arabidopsis thaliana and A. lyrate were chosen and there is no information regarding this issue in the results (page 7).
In the manuscript the names of viruses, hosts, genus, families should be written according to ICTV rules
Discussion – the epidemiological aspect of the findings was rather omitted, it will be worth to mention about potential sources of infection, transmission pathways of detected viruses, meaning of mixed infection and potential threat to other plant species in case of spreading of the viruses.
Round 2
Reviewer 1 Report
Comments and Suggestions for Authors
Authors have cleared the critical points from my previous review and have defended their point of view concerning the manuscript structure, therefore I recommend the manuscript forpublication.